# Personalized Exercise Recommendation with Semantically-Grounded Knowledge Tracing

**Yilmazcan Ozyurt**[*][†]
ETH Zürich

**Tunaberk Almaci**[*]
ETH Zürich

**Stefan Feuerriegel**
Munich Center for Machine Learning & LMU Munich

**Mrinmaya Sachan**
ETH Zürich

## Abstract

We introduce **ExRec**, a general framework for personalized exercise recommendation with semantically-grounded knowledge tracing. Our method builds on the observation that existing exercise recommendation approaches simulate student performance via knowledge tracing (KT) but they often overlook two key aspects: (a) the semantic content of questions and (b) the sequential, structured progression of student learning. To address this, our ExRec presents an end-to-end pipeline, from annotating the KCs of questions and learning their semantic representations to training KT models and optimizing several reinforcement learning (RL) methods. Moreover, we improve standard Q-learning-based continuous RL methods via a tailored model-based value estimation (MVE) approach that directly leverages the components of KT model in estimating cumulative knowledge improvement. We validate the effectiveness of our ExRec using various RL methods across four real-world tasks with different educational goals in online math learning. We further show that ExRec generalizes robustly to new, unseen questions and that it produces interpretable student learning trajectories. Together, our findings highlight the promise of KT-guided RL for effective personalization in education.

## 1 Introduction

The rapid rise of online learning platforms has revolutionized education by offering students access to interactive learning materials and exercises [3, 29, 41, 53]. A key factor in enhancing student learning is *personalization*, where exercises are recommended based on students' evolving knowledge states [22, 77].

A central method for assessing learning progress is *knowledge tracing* (KT), which models the temporal dynamics of student learning by predicting responses to future exercises. This enables real-time monitoring of knowledge states, which are structured around fundamental skills known as knowledge concepts (KCs). Over the years, numerous KT approaches have been developed [e. g., 1, 12, 15, 20, 39, 59, 61, 63, 70, 71, 81, 83][1]. Yet, only a small number of works have actually leveraged KT for personalized exercise recommendations.

Recent methods for personalized exercise recommendation have integrated KT into reinforcement learning (RL) frameworks to simulate student behavior and learn optimal recommendation policies [4, 10, 14, 42, 75, 82]. However, these methods suffer from several key **limitations**: ① they often rely on ID-based embeddings, which neglects the semantics of questions; ② they define states as the full exercise history, making them computationally impractical for long exercise sequences; ③ the

---

[*]Equal contribution.

[†]Correspondence to Yilmazcan Ozyurt <yozyurt@ethz.ch>.

[1]For a more comprehensive overview, we refer to [2] and [68].

39th Conference on Neural Information Processing Systems (NeurIPS 2025).

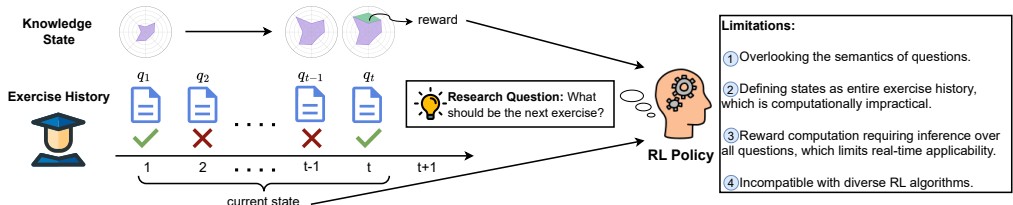

Figure 1: **Overview of standard exercise recommendation and its limitations.**

reward computation requires inference over all questions, which limits real-time applicability; and ④ they typically support only a single RL algorithm.

To address these challenges, we introduce **ExRec**, a novel framework for personalized exercise recommendation with semantically-grounded knowledge tracing. ExRec operates with minimal requirements, relying only on question content and exercise histories. It automates an end-to-end pipeline: (i) it annotates each question with solution steps and KCs, (ii) learns semantically meaningful embeddings of questions and KCs, (iii) trains KT models to simulate student behavior and calibrates them to enable direct prediction of KC-level knowledge states, and (iv) supports efficient RL by designing compact student state representations and KC-aware reward signals. In addition, we propose a model-based value estimation (MVE) approach that leverages the structure of the KT environment to guide and stabilize the Q-learning of continuous RL algorithms.

Overall, our **contributions** are as follows:[2]

- We integrate automated KC annotation and contrastive learning modules to learn rich semantic representations of questions for downstream exercise recommendation.

- We design a *compact state representation* for students, which eliminates the need to process full exercise histories, and *efficient knowledge state computation*, eliminating the need to run inference over a large set of questions.

- We introduce a model-based value estimation technique that directly leverages the KT model for computing value functions to improve Q-learning-based continuous RL methods for our task.

- We validate ExRec through extensive experiments using various RL algorithms to demonstrate its effectiveness in improving knowledge state estimation and exercise recommendation quality.

## 2 Related Work

**Knowledge tracing (KT)** models the temporal dynamics of students' learning process and predicts the response of the student to the next exercise [20]. Over the years, numerous KT methods have been proposed [1, 13, 15, 21, 28, 30, 35, 38, 40, 43, 44, 47, 49, 50, 52, 56–60, 63, 67, 69, 70, 79, 78, 80, 81, 83]. Yet, only a few studies have explored KT for optimizing exercise recommendations.

**Exercise recommendation** has been widely studied in conjunction with KT[3], primarily as a means to simulate student behavior and inform reinforcement learning (RL)-based policies for personalized sequencing. However, existing methods suffer from key limitations. Early approaches [4, 36, 42] rely on handcrafted rewards and treat questions as predefined mappings to KCs, preventing them from leveraging (dis)similarity between exercises. Similarly, methods such as TGKT-RL [14] require a predefined question-KC graph, which is often unavailable, while others operate solely at the KC level without considering individual exercises [10, 82]. RCL4ER [75] does not consider KCs of questions and learns embeddings for each question ID.

The existing works [4, 10, 14, 36, 42, 75, 82] face further common challenges. They do not effectively leverage question semantics, often relying on ID-based embeddings or simple heuristics. These methods define states as the entire exercise history, making them computationally impractical for long exercise sequences. Moreover, reward calculation in these methods requires inference over the full question set, making real-time decision-making inefficient. Finally, they support only a single RL method for exercise recommendation.

We address these limitations by ① learning rich semantic representations of questions, additionally allowing generalization to unseen exercises, ② modeling compact student states, which eliminates

---

[2]Code and trained models are provided in https://github.com/oezyurty/ExRec .

[3]We discuss the exercise recommendation methods without KT in Appendix A.

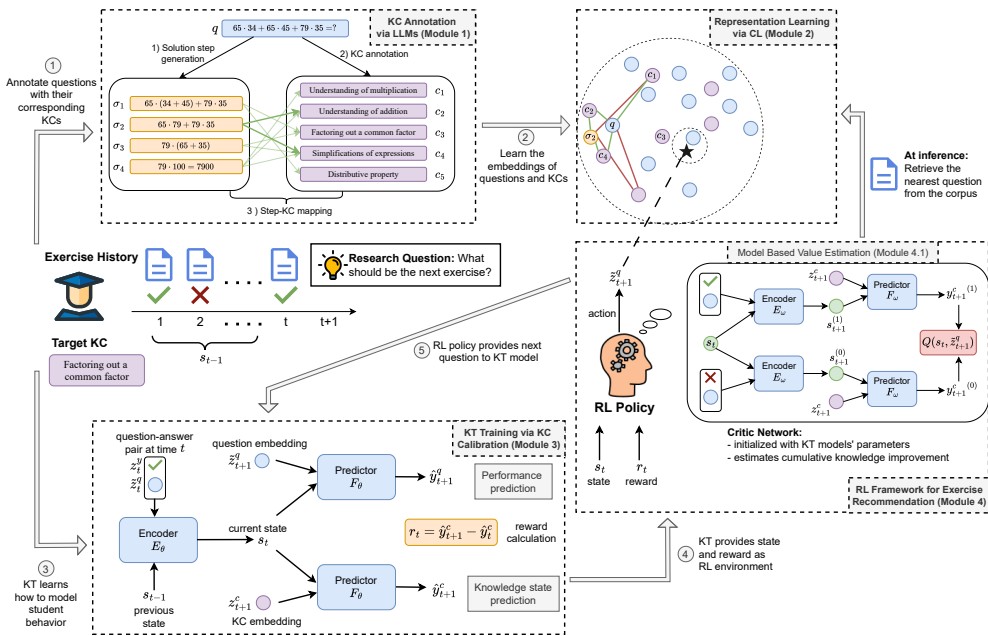

Figure 2: **Overview of ExRec framework.** Numbered gray arrows show how modules interact.

the need for full exercise histories, ③ computing knowledge states directly, avoiding exhaustive inference over question sets, and ④ supporting a broad range of RL algorithms, including both discrete and continuous action spaces. Unlike prior works, we make our entire pipeline open-source to enable researchers to instantly build and test new exercise recommenders within our framework.

## 3 Preliminaries

**Knowledge tracing (KT).** KT aims to predict a student's performance on the next exercise based on their learning history [63, 70, 83]. A student's exercise history over $t$ time steps is represented as $\{e_i\}_{i=1}^t$, where each exercise $e_i$ consists of a question $q_i \in \mathbb{Q}$, of associated knowledge concepts (KCs) $\{c_{i,j}\}_{j=1}^{N_{q_i}}$ with $c_{i,j} \in \mathbb{C}$, and of the student's binary response $y_i \in \{0, 1\}$. The KT model $G_\theta$ predicts the probability of a correct response for the next question, i.e., $\hat{y}_{t+1} = G_\theta\big(q_{t+1}, \{c_{t+1,j}\}_{j=1}^{N_{q_{t+1}}}, \{e_i\}_{i=1}^t\big)$.

**Exercise recommendation via RL.** KT models can serve as RL environments to simulate student learning behavior, formulated as a Markov decision process (MDP) $(\mathcal{S}, \mathcal{A}, P, R, \gamma)$ [4, 10, 75]. The state $s_t \in \mathcal{S}$ represents the exercise history $s_t = \{e_i\}_{i=1}^t$, and the action $a_t \in \mathcal{A}$ corresponds to selecting the next question $q_{t+1} \in \mathbb{Q}$. The transition function $P$ determines the next state based on the student's response: $P(s_{t+1} \mid s_t, q_{t+1}) : \mathcal{S} \times \mathcal{A} \times \mathcal{S} \to [0, 1]$. The reward $R$ measures the improvement in the student's knowledge state after solving $q_{t+1}$. Finally, $\gamma$ is the discount factor. In reward calculation, existing methods [e. g., 4, 10] compute knowledge states by running $G_\theta$ over all questions from a given KC at each step, leading to high computational overhead.

## 4 ExRec Framework

Our complete ExRec framework has four modules (see Fig. 2). **(1)** The first module takes the question content and annotates its solution steps and associated knowledge concepts (KCs) in an automated manner. **(2)** The second module learns the semantics of questions using the solution steps and KCs via a tailored contrastive learning objective. **(3)** The third module trains a KT model using question semantics and student histories. To better simulate student performance in an RL setup, we **(i)** train the KT model for next-exercise performance prediction, and **(ii)** calibrate its knowledge state predictions over each KC. **(4)** The final module involves an RL framework for exercise recommendation, using the calibrated KT model as the environment. For Q-learning-based continuous RL methods, we further improve training with a model-based value estimation (MVE) approach, which incorporates the KT model itself to simulate future interactions and is beneficial for estimating the policy value.

### 4.1 KC Annotation via LLMs (Module 1)

Existing works on modeling learning progress [e. g., 16, 51, 73] typically require datasets containing thousands of questions with manually-annotated KCs. However, this annotation process is expensive and error-prone, as it requires domain experts to assign KCs from hundreds of categories while maintaining consistency [8, 18]. In contrast, our framework circumvents the need for manual labeling, as described below.[4]

Informed by the recent literature [58], our ExRec framework automates KC annotation using a large language model (LLM) through a three-step process. In doing so, we further instruct the LLM to align the KC annotation with the *Common Core State Standards for Mathematics*[5] to improve the consistency in the KC annotation and to support uptake of our framework in education.[6]

**(i) Solution step generation.** Given a question $q \in \mathbb{Q}$, the LLM generates a step-by-step solution $\Sigma = \{\sigma_1, \sigma_2, \ldots, \sigma_n\}$ using chain-of-thought prompting. Each step $\sigma_k$ is sampled sequentially via

$$\sigma_k \sim P_\phi(\sigma_k \mid q, \sigma_1, \ldots, \sigma_{k-1}), \tag{1}$$

where $P_\phi$ is the LLM's probability distribution conditioned on the question and the previous steps.

**(ii) KC annotation.** The LLM then assigns a set of knowledge concepts $C = \{c_1, c_2, \ldots, c_m\}$ based on the question and its solution steps, sampled iteratively via

$$c_j \sim P_\phi(c_j \mid q, \Sigma, c_1, \ldots, c_{j-1}). \tag{2}$$

**(iii) Solution step–KC mapping.** Not all KCs are applied in every solution step. To establish a structured mapping, the LLM generates pairs iteratively via

$$(\sigma_k, c_j) \sim P_\phi(\sigma_k, c_j \mid q, \Sigma, C, \mathcal{M}'), \tag{3}$$

where $\mathcal{M}'$ represents previously assigned mappings. The final mapping is

$$\mathcal{M} = \{(\sigma_k, c_j) \mid \sigma_k \in \Sigma \wedge c_j \in C_k\}, \tag{4}$$

where $C_k$ is the set of KCs the solution step $\sigma_k$ is mapped to. The mapping then serves as the foundation for representation learning in the next module.

### 4.2 Representation Learning via Contrastive Learning (Module 2)

Our ExRec framework employs contrastive learning (CL) to generate semantically meaningful embeddings for questions, solution steps, and knowledge concepts (KCs). Instead of using general-purpose embeddings, CL explicitly aligns questions and solution steps with their associated KCs while mitigating false negatives.

**Embedding generation.** Recall that, formally, after the annotation steps from the previous module, a question $q$ has $N$ solution steps $\{\sigma_k\}_{k=1}^N$ and $M$ knowledge concepts $\{c_j\}_{j=1}^M$. We first encode the question content, its solution steps, and KCs using a learnable (LLM) encoder $E_\psi(\cdot)$:

$$z^q = E_\psi(q), \quad z_k^\sigma = E_\psi(\sigma_k), \quad z_j^c = E_\psi(c_j). \tag{5}$$

These embeddings are later optimized via contrastive learning.

**False negative elimination via clustering.** Some KCs describe the same underlying skill using slightly different wording (e.g., *interpreting a bar chart* vs. *reading information from a bar graph*). Although semantically equivalent, these variants may yield different embeddings and be mistakenly treated as negatives. To prevent this, we pre-cluster KCs using a semantic clustering function $A(\cdot)$, thereby ensuring that KCs within the same cluster are not considered as negatives:

$$A(c_j) = A(c_{j'}) \Rightarrow c_{j'} \notin \mathcal{N}(c_j), \tag{6}$$

where $\mathcal{N}(c_j)$ denotes the set of valid negative KCs for contrastive learning.

**Contrastive learning for questions.** Since each question $q$ is associated with multiple KCs $C = \{c_1, c_2, \ldots, c_m\}$, its embedding should be closer to its relevant KCs while being pushed apart from

---

[4]In our experiments, we observed that existing KC annotations are often noisy and that LLM-based annotation yields higher-quality and more coherent results. This observation echoes the findings of recent works [55, 58]. Nevertheless, we provide an in-depth analysis in Appendix G. Therein, we illustrate a representative example where our framework provides consistently higher-quality KC annotations than the original dataset.

[5]https://www.thecorestandards.org/Math/

[6]We also experimented with Bloom's Taxonomy [5] but found that it was not equally helpful for our setting.

irrelevant ones. For this, we use the loss

$$L_q(z^q, z_j^c) = -\log \frac{\exp\left(\mathrm{sim}(z^q, z_j^c)/\tau\right)}{\exp\left(\mathrm{sim}(z^q, z_j^c)/\tau\right) + \sum\limits_{c_{j'} \in \mathcal{N}(c_j)} \exp\left(\mathrm{sim}(z^q, z_{j'}^c)/\tau\right)}, \qquad (7)$$

where $\mathrm{sim}(\cdot, \cdot)$ is the cosine similarity and $\tau$ is a temperature parameter.

**Contrastive learning for solution steps.** Similarly, each solution step $\sigma_k$ should be aligned with the KCs it practices. We thus use the loss

$$L_s(z_k^\sigma, z_j^c) = -\log \frac{\exp\left(\mathrm{sim}(z_k^\sigma, z_j^c)/\tau\right)}{\exp\left(\mathrm{sim}(z_k^\sigma, z_j^c)/\tau\right) + \sum\limits_{c_{j'} \in \mathcal{N}(c_j)} \exp\left(\mathrm{sim}(z_k^\sigma, z_{j'}^c)/\tau\right)}. \qquad (8)$$

**Final objective.** Our framework jointly optimizes the embeddings of questions and solution steps:

$$L = \frac{1}{M} \sum_{j=1}^{M} L_q(z_q, z_j^c) + \frac{1}{N} \sum_{k=1}^{N} \frac{1}{|C_k|} \sum_{c_j \in C_k} L_s(z_k^\sigma, z_j^c). \qquad (9)$$

The learned embeddings enrich the input space of KT models with semantically meaningful representations, enabling us to accurately capture the temporal dynamics of students' learning process.

### 4.3 KT Training with KC Calibration (Module 3)

Finally, we replace the standard, randomly initialized question embeddings of a KT model with our learned representations. Specifically, for each question $q_i$ in the exercise history, we define

$$\tilde{z}_i^q = \frac{z_i^q + \tilde{z}_i^\sigma}{2}, \quad \text{with} \quad \tilde{z}_i^\sigma = \frac{1}{N_i} \sum_{k=1}^{N_i} z_{ik}^\sigma, \qquad (10)$$

where we average the question embedding $z_i^q$ with the (mean-pooled) solution-step embeddings $\tilde{z}_i^\sigma$. We then feed these vectors directly into an existing KT model.

**Input preparation to KT.** For a student's history of $t$ exercises $\{e_i\}_{i=1}^t$, each exercise $e_i$ is now modeled as a tuple, i. e., $e_i = (\tilde{z}_i^q, z_i^y)$, where $z_i^y$ is the embedding of student's binary response (incorrect/correct) to the question $q_i$ and it is learned during the KT training. Note that our exercise modeling for $e_i$ is different from a typical KT formulation (see Sec. 3). As the recent research [58] suggests, the integration of question embeddings enhances predictive accuracy, as the KT model benefits from explicitly encoded question semantics rather than relying on question/KC IDs alone.

**State representation.** The state $s_t$ of a KT model equivalently represents the state of the RL environment in our ExRec framework (details in Sec. 4.4). Following prior literature [50, 59, 63, 81], the history of exercises $\{e_i\}_{i=1}^t$ can be encoded into a latent state $s_t$ via a state encoder $E_\theta$ as

$$\textbf{(a) } s_t = E_\theta(\{\tilde{z}_i^q, z_i^y\}_{i=1}^t) \quad \text{or} \quad \textbf{(b) } s_t = E_\theta(s_{t-1}, \tilde{z}_t^q, z_t^y), \qquad (11)$$

where **(a)** represents a flexible way of encoding the entire sequence into a latent representation and **(b)** represents the recurrent way of encoding the sequence.

In our framework, we use **(b)** to compute a compact state representation $s_{t+1}$ from the current state $s_t$. The benefit is that it avoids the need to keep the entire exercise history in the replay buffer.

**KT training.** The KT model predicts the performance of the student on the next question $q_{t+1}$ via a classifier $F_\theta$ based on the current state $s_t$ via

$$\hat{y}_{t+1}^q = F_\theta(s_t, \tilde{z}_{t+1}^q). \qquad (12)$$

Overall, during the KT training, $E_\theta$ and $F_\theta$ are jointly trained on the entire history of exercises via a binary cross entropy loss

$$L_{\mathrm{pred}} = -\sum_t \left(y_t^q \log \hat{y}_t^q + (1 - y_t^q) \log(1 - \hat{y}_t^q)\right), \qquad (13)$$

where $y_t^q \in \{0, 1\}$ is the ground-truth response of the student to the given question.

**Calibration of the KT model for knowledge state prediction on any KC.** Aligned with the education literature [19, 20, 32], we define the knowledge state of a student for a particular KC as the expected performance of the student on all questions from the same KC. Specifically, for a particular KC $c \in \mathbb{C}$ at time $t$, we formalize the knowledge state of a student as

$$y_t^c = \mathbb{E}_{q \sim \mathbb{Q}(c)}[y_t^q] \approx \frac{1}{|\mathbb{Q}(c)|} \sum_{q \in \mathbb{Q}(c)} \hat{y}_t^q, \qquad (14)$$

where $\mathbb{Q}(c)$ is the set of all questions from the KC $c$.

Of note, the above approach to computing a knowledge state $y_t^c$ requires multiple inferences over a set of questions, which becomes computationally challenging in real-time, especially for a large corpus. We are the *first* to address this challenge by allowing the KT model to directly predict the knowledge state at the inference time.

To speed up knowledge state prediction, we proceed as follows. Recall that, with the introduction of our representation learning module (Sec. 4.2), both question embeddings (i.e., $z^q$) and KC embeddings (i.e., $z^c$) are already in the same embedding space. Therefore, we can formulate the knowledge state prediction as a prediction task over the KC embeddings. However, applying this prediction directly over the KC embeddings yields suboptimal results in practice (see AppendixD). To achieve the desired performance, we further calibrate the KT model and bring its prediction over a KC embedding closer to the knowledge state estimated in Eq. 14. Specifically, for each student and each time step, we sample a KC $c \in \mathbb{C}$ at uniformly random, and predict the knowledge state via

$$\hat{y}_{t+1}^c = F_\theta(s_t, z_{t+1}^c), \tag{15}$$

where $s_t$ is calculated as earlier in Eq. 11. Then, we define our knowledge state prediction loss as

$$L_{\text{KC}} = -\sum_t \left( y_t^c \log \hat{y}_t^c + (1 - y_t^c) \log(1 - \hat{y}_t^c) \right), \tag{16}$$

and define our calibration loss as

$$L_{\text{calib}} = L_{\text{pred}} + L_{\text{KC}}, \tag{17}$$

where we keep original prediction loss $L_{\text{pred}}$ to retain the prediction performance of the KT model.

## 4.4 RL Framework for Exercise Recommendation (Module 4)

The final module of our ExRec formulates exercise recommendation as a reinforcement learning (RL) task, where the calibrated KT model serves as the RL environment to simulate student learning behavior. Here, we define a Markov decision process (MDP) based on the learned representations and knowledge state predictions from our KT model, which enables seamless integration with any standard RL algorithm.

**MDP formulation.** We define the RL problem as an MDP $\mathcal{M} = (\mathcal{S}, \mathcal{A}, P, R, \gamma)$, where:

- **State space** ($\mathcal{S}$): The student's state at time $t$, denoted as $s_t$, is represented by the KT model's compact student state, i.e., $s_t = E_\theta(s_{t-1}, \tilde{z}_t^q, z_t^y)$, where $\tilde{z}_t^q$ is the question embedding and $z_t^y$ represents the embedding of student's past response.

- **Action space** ($\mathcal{A}$): The RL agent selects the next exercise $q_{t+1}$, represented by its embedding $\tilde{z}_{t+1}^q \in R^d$. The action space is originally ***continuous***, and exercises are retrieved via semantic similarity in the learned representation space at the test time. Our ExRec framework additionally allows RL agents with ***discrete*** action, whose output represents the question ID $q_{t+1} \in \mathbb{Q}$. This question ID is then mapped to its original embedding $\tilde{z}_{t+1}^q$ as an action for our RL environment.[7]

- **Transition dynamics** ($P$): The environment transition is governed by the KT model, which updates the student's state $s_{t+1}$ based on their response to the selected question. Specifically, the transition probability is defined as $P(s_{t+1} \mid s_t, \tilde{z}_{t+1}^q) : \mathcal{S} \times \mathcal{A} \times \mathcal{S} \to [0, 1]$, where the next state $s_{t+1}$ is determined by the student's correctness on the recommended question $q_{t+1}$. Given the KT model's predicted probability of a correct response $\hat{y}_{t+1}^q = P(y_{t+1} = 1 \mid s_t, \tilde{z}_{t+1}^q)$, the next state follows a probabilistic update[8]:

$$s_{t+1} = \begin{cases} E_\theta(s_t, \tilde{z}_{t+1}^q, z_{t+1}^{y=1}), & \text{with probability } \hat{y}_{t+1}^q, \\ E_\theta(s_t, \tilde{z}_{t+1}^q, z_{t+1}^{y=0}), & \text{with probability } 1 - \hat{y}_{t+1}^q. \end{cases} \tag{18}$$

All other next states have zero probability. Therefore, the transition mechanism is aligned with the temporal dynamics of student's learning process, as modeled by the calibrated KT model.

- **Reward function** ($R$): The reward reflects the improvement in the knowledge state. Given a KC $c$, we define the reward as the change in predicted knowledge state, i.e.,

$$r_t = \hat{y}_{t+1}^c - \hat{y}_t^c, \tag{19}$$

where $\hat{y}_t^c$ is the student's predicted knowledge state for $c$ at time $t$, computed directly via the calibrated KT model.

- **Discount factor** ($\gamma$): Controls the trade-off between short-term and long-term learning gains.

---

[7]Regardless of action space being continuous or discrete, the RL environment, i.e., the calibrated KT model, simulates the student behavior the same way as it only processes the question embeddings.

[8]For clarity, we distinguish incorrect/correct binary response embeddings as $z_t^{y=0}$ and $z_t^{y=1}$, respectively.

**Integrating KT into RL.** Our approach natively integrates KT within the RL framework. This is enabled by the calibrated KT model, which provides: **(i)** Compact student state representation without requiring full student history encoding, and **(ii)** direct knowledge state estimation instead of relying on indirect performance proxies. Thanks to this, various RL algorithms (e.g., PPO [66], TD3 [27], SAC [31]) can be used in a seamless manner for optimizing exercise selection policies.

### 4.4.1  Model-Based Value Estimation

For continuous-action RL algorithms with Q-learning [e. g., 27, 31, 46], we optimize exercise selection by estimating the expected future knowledge improvement using a critic network. Instead of training a randomly initialized Q-network from scratch, we leverage the "full access to our environment", i.e., the KT model, to design a model-based value estimation method [26].

**Bellman optimality in our RL environment.** In standard Q-learning, the optimal action-value function satisfies the Bellman equation [7]:

$$Q^*(s_t, \tilde{z}_{t+1}^q) = \mathbb{E}\big[r_t + \gamma \max_{\tilde{z}_{t+2}^q} Q^*(s_{t+1}, \tilde{z}_{t+2}^q) \mid s_t, \tilde{z}_{t+1}^q\big]. \tag{20}$$

Here, $Q^*(s_t, \tilde{z}_{t+1}^q)$ represents the expected cumulative knowledge improvement if the RL agent selects question $q_{t+1}$ in state $s_t$.

**Model-based critic design.** Rather than learning $Q(s_t, \tilde{z}_{t+1}^q)$ from trial-and-error interactions, we exploit the KT model's structure to directly estimate student progression. We construct a novel critic network initialized with components from the pre-trained KT model:

- The state transition function $E_\omega$ is initialized as $E_\theta$, the recurrent knowledge tracing module from our KT model.

- The value prediction function $F_\omega$ is initialized as $F_\theta$, the calibrated KT prediction module.

**Estimating the value function.** Given the current student state $s_t$ and selected question $z_{t+1}^q$, we compute the expected future knowledge improvement based on two possible responses:

$$s_{t+1}^{(1)} = E_\omega(s_t, \tilde{z}_{t+1}^q, z_{t+1}^{y=1}), \quad s_{t+1}^{(0)} = E_\omega(s_t, \tilde{z}_{t+1}^q, z_{t+1}^{y=0}), \tag{21}$$

where $s_{t+1}^{(1)}$ and $s_{t+1}^{(0)}$ are the next states if the student answers correctly or incorrectly, respectively. The predicted accumulated knowledge state for the targeted KC in each scenario is obtained via

$$y_{t+1}^c{}^{(1)} = F_\omega(s_{t+1}^{(1)}, z^c), \quad y_{t+1}^c{}^{(0)} = F_\omega(s_{t+1}^{(0)}, z^c). \tag{22}$$

**Final value computation.** The final Q-value is computed as the expected knowledge improvement over the current knowledge state, weighted by the KT model's response prediction $\hat{y}_{t+1}^q$:

$$Q(s_t, \tilde{z}_{t+1}^q) = \hat{y}_{t+1}^q \cdot y_{t+1}^c{}^{(1)} + (1 - \hat{y}_{t+1}^q) \cdot y_{t+1}^c{}^{(0)} - \hat{y}_t^c, \tag{23}$$

where $\hat{y}_t^c$ is the knowledge state of the student *before* answering the question, acquired by the KT model. This formulation ensures that the value estimate aligns with the student's expected learning progression by directly leveraging the KT model rather than relying solely on learned Q-values.

By integrating this model-based critic, our RL agent benefits from **(i)** accurate, structured knowledge estimation, which reduces the need for excessive environment interactions, **(ii)** efficient policy learning, as the Q-function is informed by the calibrated KT model, and **(iii)** seamless adaptation to new students and exercises, as the critic inherently captures the student's evolving knowledge state.

## 5  Experimental Setup

**Dataset.** We use the XES3G5M dataset [51], a large-scale KT benchmark with high-quality math questions. It contains 7,652 unique questions and 5.5M interactions from 18,066 students. As the original questions are in Chinese, we have translated them into English. See Appendix B for details.

**RL environment.** To ensure realistic student behavior, we initialize the RL environment by sampling a student and encoding their first 100 exercises. The 100th latent state serves as the initial state of the RL agent, ensuring a sufficient number of knowledge states across different environments to avoid cold-start problems. Following initialization, our RL agents interact with the environment for a fixed horizon of 10 steps, where each step corresponds to an exercise recommendation. In evaluation, we compare RL algorithms across 2048 students, i. e., environments, from the test set of the dataset.

**Non-RL Baselines.** To assess the extent to which RL agents learn meaningful exercise recommendation policies, we first implement two non-RL baselines: (i) a *random policy*, which recommends

exercises uniformly at random from the existing question corpus, and (ii) *historical data*, where the knowledge state evolves based on the actual student responses.[9]

**RL algorithms.** To evaluate the effectiveness of our ExRec framework, we integrate a broad range of RL algorithms for personalized exercise recommendation[10]. We start with (i) *continuous state-action methods*, which include both value-based approaches (DDPG [46], SAC [31], and TD3 [27]) and policy-based algorithms (TRPO [65] and PPO [66]). We then consider (ii) *discrete action methods* such as Discrete SAC [17], C51 [6], Rainbow [34], and DQN [54]. For DDPG, SAC, and TD3, we also provide variants with model-based value estimation (denoted as "w/ MVE"), which utilize our calibrated KT model to improve value estimation and long-term reward propagation.

**Implementation.** Appendix C provides the details of our ExRec, including model configurations, training procedures, runtime, and hyperparameters, to ensure reproducibility and fair comparison.

**Ablation studies.** We also evaluate the performance of the earlier modules (e. g., KC annotation, representation learning, and KT training and calibration) in Appendix D.

### 5.1 Evaluation Tasks

We evaluate our ExRec framework across four real-world tasks, each designed to reflect a different educational objective. The first task aligns with prior literature [4, 10, 14], while the remaining three are novel and address practical use cases that have been previously underexplored. A brief summary of each task is provided below, with full formulations and scoring details in Appendix E.

**Task 1: Global knowledge improvement.** We first follow the earlier literature and perform the standard task focusing on holistic student learning by optimizing improvements across all KCs. Instead of targeting a single concept, we define the reward as the aggregate change in the student's knowledge state, averaged over all KCs. Compared to the existing works, we achieve this efficiently without running inference over the full question set, thanks to our calibrated KT model.

**Task 2: Knowledge improvement in practiced KC.** We simulate a scenario where the student is focused on mastering a specific KC. For each student, we identify the most frequent KC in their last 10 exercises and set it as the target. This allows the RL agent to support ongoing learning by recommending conceptually aligned questions.

**Task 3: Knowledge improvement in upcoming KC.** To emulate curriculum progression, we compute a transition matrix between recently practiced and upcoming KCs across students. The next target KC is sampled from this distribution, enabling the RL agent to recommend questions that align with natural learning paths observed in the data.

**Task 4: Knowledge improvement in weakest KC.** At each step, we adaptively target the KC with the lowest estimated knowledge among all KCs. This way, the RL agent focuses on the student's weakest areas and personalizes learning toward balanced conceptual understanding. For evaluation, we compute the average knowledge improvement in these targeted KCs.

## 6 Results

Figure 3 shows the performance of each method as a percentage of the maximum achievable improvement in the knowledge state. We find the following: **(1)** Across all tasks, non-RL baselines yield marginal or even negative gains, which highlights the need for tailored recommendation policies. **(2)** Discrete-action methods generally perform well in simpler tasks (1–3), which is due to their ability to optimize toward a static target. **(3)** Among continuous state-action RL methods, value-based approaches (DDPG, TD3, SAC) consistently outperform policy-based ones (TRPO, PPO).**(4)** Our model-based value estimation (w/ MVE) proves especially effective, consistently boosting the performance of continuous value-based RL methods across all tasks. **(5)** Particularly at task 4, which is substantially more challenging due to that the target KC may change at each step per student, our MVE approach boosts the performance of continuous value-based RL methods beyond the level of discrete methods. *Takeaway:* Our ExRec shows the importance of tailored exercise recommendation policies. It also demonstrates how fully leveraging the KT environment, via our model-based value estimation, can lead to large improvements in RL performance for educational settings.

---

[9]Unlike all other methods, historical data does not rely on the KT model to sample student responses, as ground-truth interactions are available.

[10]We implement RL algorithms using the Tianshou library [74].

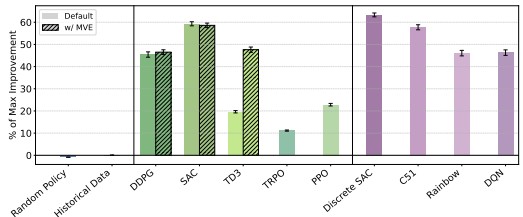

(a) Task 1: Global Knowledge Improvement.

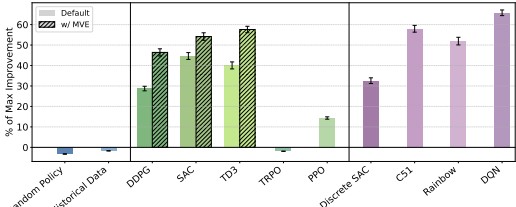
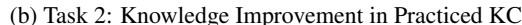

(b) Task 2: Knowledge Improvement in Practiced KC.

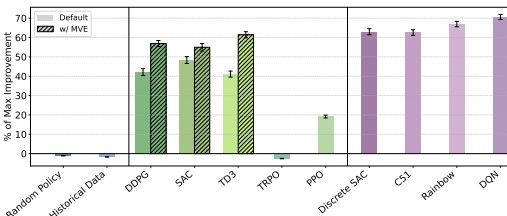

(c) Task 3: Knowledge Improvement in Upcoming KC.

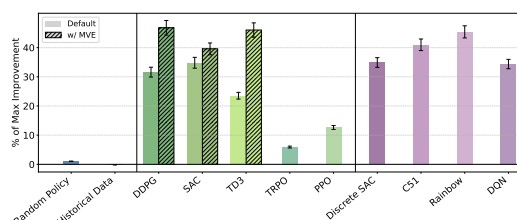

(d) Task 4: Knowledge Improvement in Weakest KC.

Figure 3: **Knowledge improvements** across four tasks, averaged over 2048 students in the test set. Our framework supports a range of RL algorithms and enables extensive comparison among methods.

## 6.1 Use Case: Extending the Question Corpus

To test whether our framework generalizes to new, unseen questions, we extend the original question corpus by generating three times more questions using GPT-4o (see Appendix H for prompt details and examples). Each generation introduces additional KCs while preserving the conceptual grounding of the original question. As shown in Figure 4, default models generally experience a drop in performance under the extended corpus. In contrast, models augmented with our model-based value estimation (w/ MVE) are robust and, in many cases, even improve student knowledge more effectively by leveraging the broader question set.

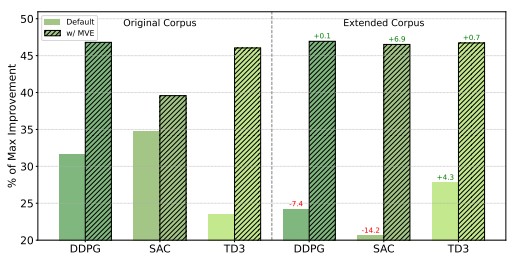

Figure 4: **Results of extending question corpus.**

## 6.2 Use Case: Visualizing Conceptual Growth

We visualize how our framework supports learning trajectories. We use a random student from Task 4 for illustration. Figure 5 shows how the student's knowledge states evolve across steps.[11] Here, we compare a value-based continuous-action model (DDPG) and its improved versions—w/ MVE and w/ MVE + extended corpus—against a random policy. **(1)** The random policy fails to produce meaningful gains, as it recommends exercises without addressing the weaknesses of the student. **(2)** DDPG achieves moderate improvements, underscoring the benefit of learning tailored policies through our framework. **(3)** Our MVE approach leads to significantly larger gains. **(4)** Further improvements are observed when training on the extended corpus, which broadens the action space while remaining compatible with our framework. **(5)** Notably, while vanilla DDPG may take several steps to address the same KC (e.g., KC 1326), our enhanced variants treat KCs more efficiently by adapting quickly, which enables focusing on other weak concepts.

## 7 Discussion

ExRec addresses limitations of earlier methods for personalized exercise recommendations by introducing semantically grounded question representations, compact student state modeling, and efficient knowledge state estimation for reward computation. As a result, we demonstrate that the personalized exercise recommendation benefits greatly from modeling both the semantics of exercises and the structure of student learning. Further, our model-based value estimation (MVE) consistently improves the performance of Q-learning-based continuous RL methods, particularly in settings like

---

[11]The knowledge state evolution for a wider set of RL policies can be found in Appendix I.

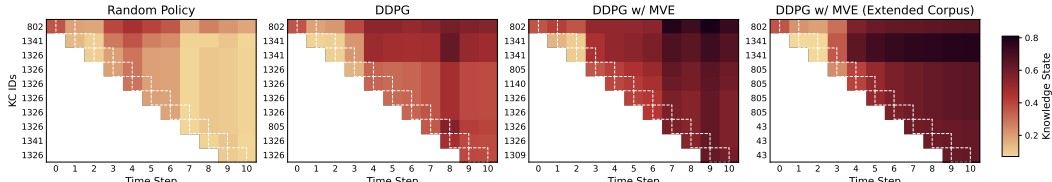

Figure 5: **Knowledge state evolution** for a single student across different policies (Task 4). At each step, the weakest KC is targeted, and the corresponding knowledge trajectory is shown. White dashed boxes mark changes after each recommendation. A KC may appear more than once if it remains the weakest. KC IDs are for visualization only; our framework uses semantic KC embeddings, not IDs.

targeting the weakest KC, where the target KC shifts dynamically over time. Finally, our framework generalizes to unseen exercises and supports fine-grained analyses of evolving student knowledge. As ExRec relies on KT models as environment and semantic embeddings as question representations, future work could explore stronger KT models or difficulty-aware question representations for even richer personalization. Together, these findings suggest that leveraging KT as an environment opens new avenues for scalable personalization in education.

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

# A    Alternative Approaches to Exercise Recommendation

Various works in exercise recommendation have explored a range of strategies, often without simulating student learning trajectories. Several approaches rely on static, rule-based logic, recommending knowledge concepts (KCs) based on manually defined prerequisite graphs [33], proxy objectives like novelty [76], or instructor-specified difficulty levels [62], rather than learning an adaptive policy. Other methods frame recommendation as a supervised learning problem, optimizing for immediate performance or success [37, 45] instead of prioritizing long-term knowledge improvement. A common thread in these works is the reliance on hand-crafted features or ID-based representations, which overlooks the semantic content of questions and KCs [33, 37, 45], with some systems providing decision-support dashboards for instructors rather than a fully automated recommendation policy [72]. These limitations motivate more dynamic approaches that can model the evolution of student knowledge over time.

# B   Dataset

The XES3G5M dataset [51] is a large-scale benchmark for knowledge tracing, collected from a real-world online math learning platform. It comprises 18,066 student histories, totaling over 5.5 million interaction records across 7,652 unique math questions. These questions are annotated with 865 leaf-level knowledge concepts (KCs). Compared to existing KT datasets, XES3G5M provides rich auxiliary information, including: (1) full question texts and types (multiple-choice and fill-in-the-blank), (2) and the final answers of the provided questions.

For compatibility with our framework, we translated all question texts from Chinese to English using a high-quality commercial translation tool, and ensured formatting consistency through post-processing. We provide the translated dataset along with our annotations in the repository.

We note that, while the dataset provides manually annotated KCs, we found that these annotations are often low-quality (see Appendix G). Therefore, we systematically re-annotate questions using our Module 1 (KC Annotation via LLMs), resulting in improved consistency and interpretability. In the original dataset, each question is associated with 1.16 KCs on average. In comparison, our ExRec framework identifies 3 KCs for the majority of questions. This can be attributed to our framework's ability to provide more comprehensive and modular KC annotations compared to the original distributions. Fig. 6 shows the distribution of number of KCs per question.

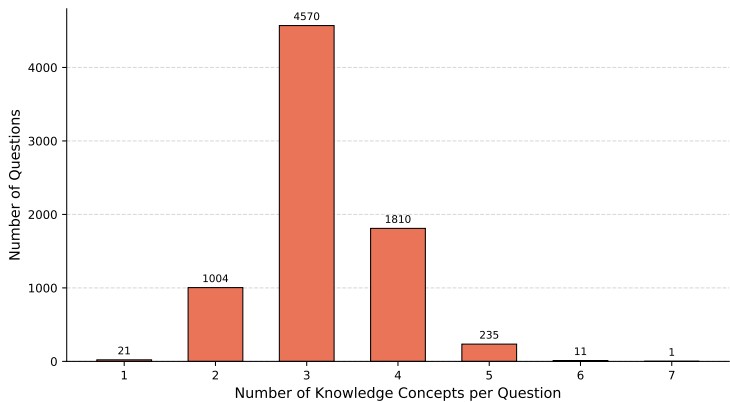

Figure 6: Distribution of number of KCs per question annotated by our framework.

Our framework produces 5,139 unique KC annotations, which are grouped into 1,377 clusters based on semantic similarity. Compared to our earlier work [58], which produced 8,378 unique KCs and 2,024 clusters, this represents a substantial improvement in the consistency of the KC annotations. We attribute this improvement to two key design choices: (i) instructing the LLM to align with the Common Core State Standards for Mathematics to ensure canonical phrasing of KCs, and (ii) prompting the model to explicitly reason about the relevance of each KC in context (see Appendix F). Fig. 7 illustrates the most frequent clusters across the dataset. We release all annotated KCs and their clustering results to support reproducibility and further research.

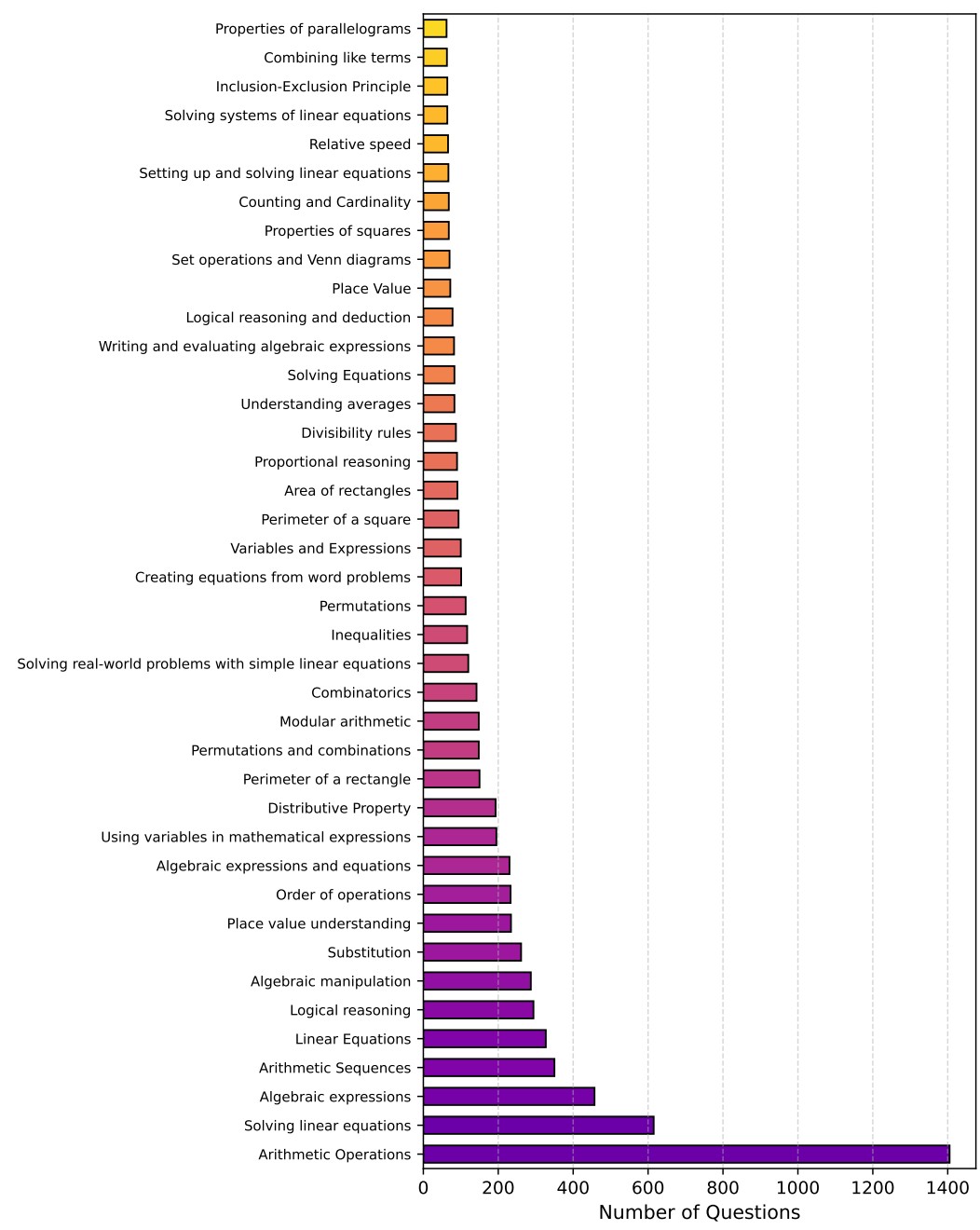

Figure 7: Most representative KCs across all questions in the dataset. The result is shown after clustering semantically similar KCs.

# C Implementation Details

**KC Annotation (Module 1).** We leverage the reasoning capabilities of OpenAI's GPT-4o[12] to automate all three stages: solution step generation, KC annotation, and solution step–KC mapping. We carefully steer model towards more consistent KC annotations across different Math questions, which is ignored in earlier works [55, 58]. For this, we specifically instruct the LLM to (1) use Common Core State Standards for Mathematics as a reference and (2) reason about why and how each KC is particularly relevant for the current question. We set the temperature to 0 for deterministic outputs. For each question in the dataset, the model is queried three times, once per stage. Across the 7,652 questions in XES3G5M, the total cost of prompting remains approximately 50 USD.

**Question Representation Learning (Module 2).** To learn semantically rich question embeddings, we adopt a multi-stage contrastive training approach similar to [58]. First, we cluster the Sentence-BERT [64] embeddings of KCs using HDBSCAN [11] to identify semantically close groups and mitigate false negatives during contrastive training. We use cosine similarity as the distance metric, and we set the minimum cluster size and minimum samples to 2.

We then fine-tune a BERT encoder [23] using a contrastive loss over question–KC and solution step–KC pairs. To differentiate the input types, we introduce three special tokens—[Q], [S], and [KC]—prepended to the question content, solution step, and KC, respectively. Their [CLS] token embeddings are used to represent the full input sequence.

We train the model for 50 epochs using a batch size of 32, a learning rate of 5e-5, dropout of 0.1, and a temperature of 0.1 in the similarity function. The training is performed on an NVIDIA A100 GPU (40GB) and completed in under 6 hours.

After training, question embeddings are computed by encoding both the question content ([Q]...) and its solution steps ([S]...) and aggregating the resulting embeddings as described in Sec. 4.3. Importantly, this inference does not require KC annotations—making it possible to directly embed newly added questions without running the LLM again.

**KT training with KC calibration (Module 3).** For an efficient state representation, we leverage a KT model that encodes the exercise history into a latent state in a recurrent manner. Specifically, we choose an LSTM architecture for the state encoder $E_\theta$, which has been used in earlier KT works [1, 49, 56, 63, 70, 79][13]. The classifier $F_\theta$ is a multi-layer perceptron. During our entire experiments, we fixed the dimensionality of state $s_t$ to 300. To ensure an equal representation with the question embeddings, we further increase the dimensionality of $s_t$ to 768 (same as the dimensionality of question embeddings) via a linear layer before feeding it into the classifier $F_\theta$.

During KC calibration of the KT model, we approximate a student's knowledge state on a given concept $c \in \mathbb{C}$ by sampling 20 relevant questions associated with $c$ from the question corpus and averaging the predicted correctness scores. To ensure stability and prevent the moving target problem, these predictions are computed using a fixed checkpoint of the KT model obtained after standard training is completed.

Both the initial KT training and the subsequent KC calibration are performed using a batch size of 512 and a learning rate of 2e-5. We used NVIDIA GeForce RTX 3090 with 24GB GPU for the training of KT models. The training takes approximately 2.5 hours on a single GPU and the inference (per question) takes 8.7 milliseconds. Of note, fast inference is the key element to ensure efficient student simulation in RL environment.

For implementation, we customize the `pyKT` library [48] to support our custom model architecture and KC-level supervision. We provide our custom KT architecture and its training details in our repository.

**RL framework for exercise recommendation (Module 4).** We integrate our trained KT model as an RL environment within the Tianshou library [74], following the OpenAI Gym API specification [9] to ensure seamless compatibility. This design allows multiple RL agents to interact with the KT-based environment for a comprehensive and flexible benchmarking of exercise recommendation policies.

---

[12]`https://platform.openai.com/docs/models/gpt-4o`

[13]We note that we do not sacrifice the accuracy by choosing an LSTM encoder instead of attention-based encoder. As found in earlier research [58], the LSTM-based KT models achieve state-of-the-art performances after incorporating the rich question representations.

Unlike the default Tianshou strategy, which creates $N$ environment instances and manages them asynchronously via multi-threading,[14] our framework supports parallel student simulations by processing all $N$ environments as a single batch on the GPU. This enables more efficient training and significantly reduces memory overhead. We release all necessary environment wrappers and code for easy integration and reproducibility.

Similar to the training of KT models, we used NVIDIA GeForce RTX 3090 with 24GB GPU for the training of RL algorithms and each training is completed under an hour. RL inference per exercise recommendation takes 78.5 milliseconds on a single GPU, which shows the effectiveness when recommending the exercise in real time.

The hyperparameter details of RL algorithms can be found in Table 1.

---

[14]`https://tianshou.org/en/stable/01_tutorials/07_cheatsheet.html`

Table 1: Hyperparameter tuning of RL algorithms.

| Method | Hyperparameter | Tuning Range |
|---|---|---|
| All methods | Question and KC Embedding Size | 768 |
| | Critic Network Up Projection Size | 1200 |
| | Critic Network Hidden Size | 300 |
| | Max Steps | 10 |
| | Reward Scale | 1000 |
| | Gamma | 0.99 |
| | Learning Rate | $[5 \cdot 10^{-5}, 1 \cdot 10^{-4}]$ |
| DDPG | Tau | [0.001, 0.01, 0.05, 0.1] |
| SAC | Tau | [0.01, 0.015, 0.02, 0.05] |
| | Alpha | [0.1, 0.2, 0.3,0.4, 0.5] |
| | Deterministic Evaluation | [True, False] |
| TD3 | Tau | [0.05, 0.005,0.01] |
| | Actor Update Frequency | [2, 4] |
| | Policy Noise | [0.1, 0.2] |
| | Noise Clip | [0.5] |
| TRPO | Actor Step Size | [0.5, 0.025] |
| | Advantage Normalization | [False] |
| | Discount Factor | [0.99, 0.995] |
| | Gae Lambda | [0.95, 0.97] |
| | Max KL | [0.01, 0.02] |
| | Optim Critic Iters | [5] |
| | Reward Normalization | [False] |
| | Repeat Per Update | [1] |
| PPO | Advantage Normalization | [False] |
| | Deterministic Evaluation | [False] |
| | Discount Factor | [0.99, 0.95] |
| | Entropy Coefficient | [0.01, 0.05] |
| | L-Clip | [0.1, 0.2] |
| | Gae Lambda | [0.9, 0.95] |
| | Value Loss Coefficient | [0.5, 1.0] |
| Discrete SAC | Alpha | [0.1, 0.2] |
| | Estimation Step | [1, 10] |
| | Tau | [0.005, 0.01, 0.05] |
| C51/Rainbow | Clip Loss Gradient | [True, False] |
| | Discount Factor | [0.99, 0.995] |
| | Double Network | [True, False] |
| | Num Atoms | [17] |
| | Target Update Frequency | [0, 1, 10] |
| | $V_{\max}$ | [1000] |
| | $V_{\min}$ | [−1000] |
| DQN | Clip Loss Gradient | [True, False] |
| | Discount Factor | [0.99, 0.995] |
| | Estimation Step | [1] |
| | Double DQN | [True, False] |
| | Target Update Frequency | [0,1,10] |

Other than specified parameters, we use the default values from Tianshou library.

# D   Evaluating Individual Modules of ExRec

While the main paper focuses on evaluating the effectiveness of our framework for exercise recommendation, this section provides a closer examination of the upstream modules: KC annotation, representation learning, and knowledge tracing with KC calibration. Understanding the quality and behavior of these modules is essential, as they form the foundation upon which our recommendation policy is built.

**KC Annotation (Module 1).** Our framework builds on recent advances demonstrating the viability of using large language models (LLMs) for automated knowledge concept (KC) annotation of math problems [55, 24]. Prior work shows that LLM-generated annotations not only improve downstream reasoning performance [24], but are also preferred by human annotators over expert-created labels [55]. Our approach follows the annotation pipeline of Ozyurt et al. [58], which leverages step-by-step solutions to improve both the accuracy and conceptual granularity of KC labels. Their study found that LLMs using solution steps were preferred by annotators 86.9 % of the time compared to those without steps, and were favored 95 % of the time over the original KC annotations in the XES3G5M dataset. We extend this approach by explicitly instructing the LLM to align its output with the Common Core State Standards and to reason explicitly about the relevance of each KC. This leads to substantially more consistent annotations: our framework produces 5,139 unique KC labels grouped into 1,377 semantically coherent clusters, compared to 8,378 KCs and 2,024 clusters in Ozyurt et al. [58], indicating improved coherence and reduced redundancy in the KC space.

**Question Representation Learning (Module 2).** To assess the effectiveness of our representation learning module, we evaluate how well the learned embeddings reflect the KC structure. Specifically, we measure the retrieval quality of semantically related questions via a micro-averaged F1 score at the cluster level. For each KC cluster, we randomly select a representative KC and use the LLM to compute its embedding. We then retrieve the top-$N$ nearest questions in the embedding space, where $N$ is the number of questions associated with that cluster. Retrieval performance is calculated by comparing the retrieved questions against the true cluster members, and we aggregate scores across clusters to compute the final micro-F1.

Without representation learning, the default LLM embeddings yield a micro-F1 score of 0.2305, reflecting limited alignment between questions and their associated KCs. After applying our contrastive training objective, this score rises substantially to 0.8865, indicating a dramatic improvement in semantic coherence.

Figure 8 visualizes this improvement qualitatively. On the left, questions related to the same KC are scattered across the embedding space, resulting in poor semantic grouping. On the right, representation learning produces tightly clustered embeddings for related questions, clearly highlighting the benefit of our training procedure. This clustering not only boosts retrieval accuracy but also enhances downstream modules that rely on precise semantic alignment.

**KT training with KC calibration (Module 3).** In contrast to prior KT frameworks that train question embeddings from scratch or continue updating them during KT training [48, 51, 58], our approach preserves the semantically enriched embeddings obtained in Module 2 by freezing them throughout KT training. This design ensures that both questions and KCs remain in a shared, interpretable embedding space, allowing for direct querying of KC mastery. While freezing these embeddings leads to a small drop in predictive performance (AUC decreases from 82.13 to 81.26), it still substantially outperforms standard ID-based initialization (78.33 AUC). More importantly, this setup enables interpretable mastery estimation at the KC level by querying the KT model with a KC embedding directly. To evaluate this capability, we sampled 20 questions associated with each KC and compared their average predicted mastery against the KC-query prediction. The resulting mean absolute error was 0.08. We further enhanced this alignment through our KC calibration step (see Sec.4.3), which boosts the AUC to 81.65 and reduces the KC-level MAE to 0.028. These results validate that our framework not only maintains high predictive accuracy but also introduces a scalable and interpretable mechanism to estimate student mastery over individual KCs.

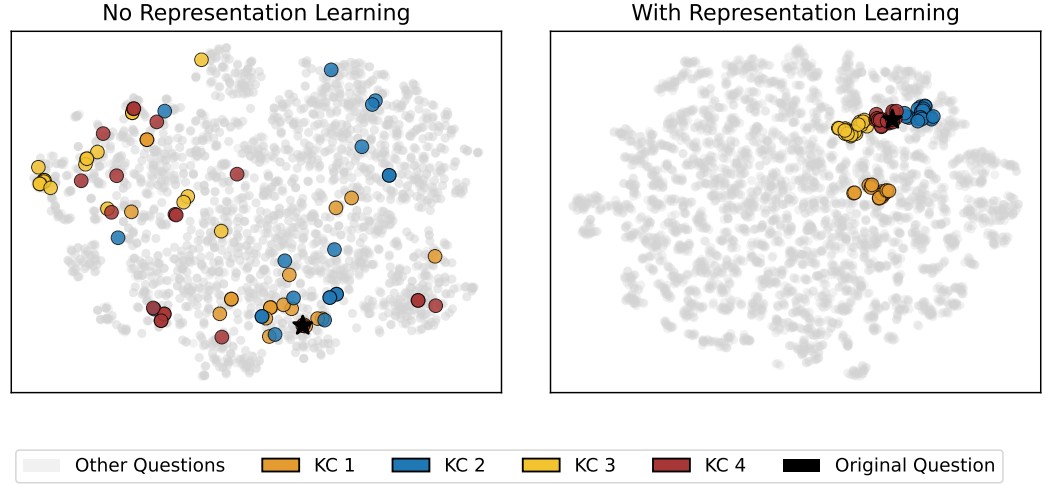

Figure 8: Effect of representation learning on question embeddings. Each color indicates a different KC cluster, and the black star marks the original query question. Our representation learning module yields tightly grouped clusters. In comparison, the clusters on the left (w/o rep. learning) are scattered.

# E    Detailed Description of Evaluation Tasks

In this section, we provide detailed formulations and motivations for the four evaluation tasks introduced in Sec. 6. Each task defines a different set of target knowledge concepts (KCs), reflecting distinct pedagogical goals. In all tasks, the RL agent interacts with a calibrated KT environment for 10 steps. While this interaction begins at the 100th exercise in the original student history, we denote this point as step $(0)$ from the RL perspective.

Let $\hat{y}^c_{(0)}$ denote the predicted knowledge state of a student for concept $c$ at the beginning of the RL episode, and $\hat{y}^c_{(10)}$ the state after 10 recommended exercises. The evaluation score is computed as the net improvement:

$$\text{Score} = \frac{1}{|T|} \sum_{c \in T} \left( \hat{y}^c_{(10)} - \hat{y}^c_{(0)} \right), \tag{24}$$

where $T$ is the set of target KCs defined for each task.

**Task 1: Global knowledge improvement.** Here, the agent is rewarded for general learning progress, with the entire concept set considered:

$$T = \mathcal{C},$$

where $\mathcal{C}$ is the set of all annotated knowledge concepts. This task evaluates broad improvement across the curriculum and is aligned with standard KT metrics. Unlike prior work, our framework allows direct inference at the concept level, avoiding costly evaluations over all questions.

**Task 2: Knowledge improvement in practiced KC.** This task models a student actively focusing on a specific concept. For each student, we extract the KCs from their last 10 exercises:

$$\mathcal{C}_{\text{recent}} = \bigcup_{i=91}^{100} \mathcal{C}_i,$$

where $\mathcal{C}_i$ is the set of KCs associated with exercise $e_i$. The most frequently occurring concept $c^* \in \mathcal{C}_{\text{recent}}$ is selected as the target: $T = \{c^*\}$.

**Task 3: Knowledge improvement in upcoming KC.** This task simulates progression in curriculum learning. We first build a KC-to-KC transition matrix $\mathcal{M}[c \to c']$ by co-occurrence statistics across student histories:

$$\mathcal{C}_{\text{before}} = \bigcup_{i=91}^{100} \mathcal{C}_i, \quad \mathcal{C}_{\text{after}} = \bigcup_{i=101}^{110} \mathcal{C}_i,$$

and for each pair $(c, c') \in \mathcal{C}_{\text{before}} \times \mathcal{C}_{\text{after}}$, we increment the matrix:

$$\mathcal{M}[c \to c'] \mathrel{+}= 1.$$

After normalization, we obtain transition probabilities:

$$P(c' \mid c) = \frac{\mathcal{M}[c \to c']}{\sum_{c''} \mathcal{M}[c \to c'']}.$$

To sample the target KC for each student, we marginalize over the set of previously encountered KCs:

$$P(c') = \sum_{c \in \mathcal{C}_{\text{before}}} P(c' \mid c) \cdot P(c),$$

where $P(c)$ is set to uniform over $\mathcal{C}_{\text{before}}$. The sampled target $c^* \sim P(c')$ defines the evaluation set $T = \{c^*\}$.

**Task 4: Knowledge improvement in weakest KC.** This task encourages the agent to target the student's weakest areas. At each step $t$, the current weakest concept is:

$$c_t^* = \arg \min_{c \in \mathcal{C}} \hat{y}^c_{(t)}.$$

Let $T = \{c_1^*, c_2^*, \ldots, c_{10}^*\}$ be the union of selected weakest concepts over the episode. The final score is computed as:

$$\text{Score} = \frac{1}{10} \sum_{i=1}^{10} \left( \hat{y}^{c_i^*}_{(10)} - \hat{y}^{c_i^*}_{(0)} \right). \tag{25}$$

This task models adaptive remediation by dynamically selecting and addressing knowledge gaps.

**Scoring details.** To evaluate performance, we measure the net change in the student's knowledge state after 10 exercise recommendations, averaged over 2048 students from the test set. Given that the dataset has an average correctness rate of 78 %, our calibrated KT model yields a mean knowledge state of 0.78 across students and KCs. Consequently, the maximum attainable improvement (on average) is +0.22, while the worst possible decline is –0.78.

Rather than reporting raw gains, we adopt a normalized metric that expresses the observed improvement as a percentage of the maximum achievable gain. For instance, if the upper bound is 0.22 and a model improves a student's knowledge by 0.15, we report 68.2 % of the maximum possible improvement. This normalization improves comparability across models and tasks. Note that while Tasks 1–3 share a common upper bound of approximately 0.22, Task 4 (due to dynamically targeting the weakest KC) permits larger gains, with upper bounds reaching up to 0.64 depending on the model.

# F Prompts for KC Annotation via LLMs

We show below the prompt templates used in Module 1 for generating solution steps, KC annotations and solution step-KC mappings. Our approach follows a system-user prompt format, which reflects how large language models are typically queried in practice.

## F.1 Solution Step Generation

**System Prompt**

```
Your task is to generate the clear and concise step by step solutions of the
    provided Math problem. Please consider the below instructions in your
    generation:

- You will be provided with the final answer. When generating the step by step
    solution, you can leverage this information piece.
- It is important that your generated step by step solution should be understandable
    as stand-alone, meaning that the student should not need to additionally check
    final answer or explanation provided.
- Your solution steps will be later used to identify the knowledge concepts
    associated at each step. Therefore, please don't write a final conclusion
    sentence as the last step, because it won't contribute to any knowledge concept.

- Don't generate any text other than the step by step solution described earlier.
- Don't divide any equation to multiple lines, i.e. an equation should start and
    finish at the same line.
- Make your step-by-step solution concise (e.g. not much verbose, and not a longer
    list than necessary) as described earlier.
- You must provide your step by step solution in a structured and concise manner in
    Solution_Steps field as a list of steps, i.e. [<step1>, ..., <stepN>] . Don't
    enumerate the steps.
- You have limited tokens, try to make each <step> as concise as possible.
- IMPORTANT: If your final answer does not match the provided final answer, provide
    one last solution step with <error>. This will help us identifying potential
    errors in your solution generation.
- IMPORTANT: Don't use any invalid character, i.e., it should be safe to call 'ast.
    literal_eval' on your response message.

Please follow the example output in json format below as a template when structuring
    your output:

{"Solution_Steps": [<step1>, <step2>, ..., <stepN>]}
```

**User Prompt Template**

```
Question: <QUESTION TEXT>
Final Answer: <FINAL ANSWER>
```

## F.2 KC Annotation

**System Prompt**

```
You will be provided with a Math question and its step by step solution. Your task
    is to provide the concise and comprehensive list of knowledge concepts (KCs) in
     Math curriculum required to correctly answer the questions.

Your task has mainly 2 phases, whose details will be provided below. Each phase has
    its own field in your json output format.

- Reasoning:
    1. Identify all the relevant KCs required to solve this problem.
    2. Justify why each KC is relevant, considering the question and solution steps.
```

```
     3. You have limited space, so please use 100 words maximum.

- List of KCs: Provide a list of unique KCs with the help of your reasoning above, i
     .e. [<KC 1>, ..., <KC M>]. Don't enumerate the KCs.
   1. Provide multiple knowledge concepts only when it is actually needed.
   2. Some questions require a figure, which you won't be provided. As the step-by-
        step solution is already provided, use your judgement to infer which
        knowledge concept(s) might be needed.
   3. For a small set of solutions, their last step(s) might be missing due to
        limited token size. Use your judgement based on your input and your ability
        to infer how the solution would conclude.
   4. Remember that knowledge concepts should be appropriate for Math curriculum.
        If annotated step-by-step solution involves advanced techniques, use your
        judgment for more simplified alternatives.

IMPORTANT NOTE: For your task, try to use the Common Core State Standards for
     Mathematics for the Knowledge Concept (KC) annotations. The reason is, we aim
     to get consistent texts for the same KCs across different questions.

Please follow the example output in json format below as a template when structuring
     your output. IMPORTANT: Don't use any invalid character, i.e., it should be
     safe to call 'ast.literal_eval' on your response message.

{"Reasoning": <Your reasoning to identify relevant KCs.>,
 "list_KCs": [<KC 1>, <KC 2>, ..., <KC M>]}
```

**User Prompt Template**

```
Question: <QUESTION TEXT>
Solution steps: <SOLUTION STEPS>
```

### F.3 Solution Step-KC Mapping

**System Prompt**

```
You will be provided with a Math question, its step by step solution and its
     associated knowledge concepts (KCs). Your task is to map each solution step
     with its associated KC(s).

- Mapping between solution steps and KCs: All solution steps and all knowledge
     concepts must be mapped, while many-to-many mapping is indeed possible.

IMPORTANT: Each solution step is already numbered from 1 to N and each knowledge
     concept is numbered from 1 to M, where M is the number of KCs you found earlier.
      For consistency, use the same ordering as your output of list of KCs. Your
     output should enumerate all solution step-knowledge concept pairs as numbers.

   1. Each solution step has to be paired.
   2. Each knowledge concept has to be paired.
   3. Map a solution step with a knowledge concept only if they are relevant.
   4. Your pairs cannot contain artificial solution steps. For instance, if there
        are 4 solution steps, the pair "5-2" is illegal.
   5. Your pairs cannot contain artificial knowledge concepts. For instance, if
        there are 3 knowledge concepts, the pair "3-5" is illegal.

IMPORTANT: For this field, you will output solution step-knowledge concept pairs in
     a comma-separated manner and in a single line. For example, if there are 4
     solution steps and 5 KCs, one potential output could be:
"1-1, 1-3, 1-5, 2-4, 3-2, 3-5, 4-2, 4-3, 4-5"

The provided example is illustrative only. Your output should reflect the actual
     mapping derived from the given question and solution.
```

```
Please follow the example output in json format below as a template when structuring
    your output. IMPORTANT: Don't use any invalid character, i.e., it should be
    safe to call `ast.literal_eval` on your response message.

{"mapping_step_KC": <solution step - knowledge concept pairs in a comma separated
    manner and in a single line.>}
```

**User Prompt Template**

```
Question: <QUESTION TEXT>
Solution steps: <SOLUTION STEPS>
KCs: <LIST OF KNOWLEDGE CONCEPTS>
```

# G Example KC Annotation

We provide an example question from the XES3G5M dataset to compare the quality of our automated KC annotation (Module 1) against the KC annotation from the original dataset.

**Question:** There is a bridge. To cross the bridge, you need to go uphill first, then walk on a level road, and finally go downhill. The total distance is 18 kilometers, and the uphill, flat, and downhill distances are equal. When someone walks across a bridge, the speed of going uphill, walking on a level road, and downhill is 2km/h, 3km/h, and 6km/h respectively. What is his average speed across the bridge in kilometers per hour?

**Solution steps:** Below is the solution steps annotated by our Module 1:

- The total distance of the bridge is 18 kilometers, and the distances for uphill, flat, and downhill are equal. Therefore, each section is $18 \div 3 = 6$ kilometers long.
- The speed going uphill is 2 km/h, so the time taken to cover the uphill section is $6 \div 2 = 3$ hours.
- The speed on the flat section is 3 km/h, so the time taken to cover the flat section is $6 \div 3 = 2$ hours.
- The speed going downhill is 6 km/h, so the time taken to cover the downhill section is $6 \div 6 = 1$ hour.
- The total time taken to cross the bridge is the sum of the times for each section: $3 + 2 + 1 = 6$ hours.
- The average speed across the bridge is the total distance divided by the total time: $18 \div 6 = 3$ km/h.

**KC annotation of the original dataset:** Single person speed change problem.

**KC annotation of our Module 1: a)** Understanding of speed, distance, and time relationship, **b)** Average speed calculation, **c)** Division, **d)** Addition.

*Takeaway:* Our ExRec provides more accurate, comprehensive, and modular KC annotations for the Math questions compared to original annotations. These annotations facilitate semantically more meaningful embeddings for questions, which are leveraged for both downstream KT models and RL policies.

# H   Artificial Question Generation for Extending the Corpus

We show below the system and user prompts used to generate diverse artificial Math questions to extend the exercise corpus. The LLM is conditioned on an original question, its step-by-step solution, and a list of annotated knowledge concepts (KCs). It is instructed to generate variations that maintain conceptual grounding while introducing structural and reasoning diversity.

**System Prompt**

```
You will be provided with a Math question, its step-by-step solution, and its
    annotated knowledge concepts (KCs). Your task is to generate conceptually
    diverse Math questions that still use the core KCs but introduce variations in
    mathematical reasoning, question framing, or constraints. You will do this 3
    times.

You can use your creativity in this task, because the variations are highly valued.
    Your generations just need to be Mathematically sound.

Please follow the detailed instructions below:

- For the first generation, you can keep the same set of KCs as in your original
    input. For the other generations, you will add one more relevant KC that you
    find appropriate. Keep in mind that, those added KCs won't accumulate over your
     last generation (i.e. you can add one more KC only over the original KCs). As
    it will be explained in the example output format, you will first decide those
    KCs at the beginning of each generation, and then you will condition your
    question on it.

- You will generate the question text based on the generated KCs you decided. Note
    that each question should modify the problem structure beyond simple rewording
    (e.g., introducing constraints, varying input conditions, changing the logical
    setup).

- Then you will generate the solution steps based on the KCs and question you
    generated. You should generate a solution that genuinely reflects the
    conceptual shift rather than being a template copy. Just like the question
    generation, your solution steps should be modified beyond simple rewording and
    they should include meaningful variations.

- You will repeat this procedure 3 times.

IMPORTANT: Ensure that your response is always a complete and well-formed JSON
    object. If you are unable to generate a proper response, provide a meaningful
    default example rather than an empty dictionary.

IMPORTANT: Diversity between your generations is highly appreciated.

Output format: You will provide a dictionary in json format. As you can see from the
    format, you will first generate the KCs, then the question content, and
    finally the solution steps. KCs and solution steps must be provided as proper
    lists in their respective fields. In the below example output format, M and N
    refer to number of KCs and solution step, which of course may differ across
    different generations. Note that generation ids are 0-indexed, i.e., from 0 to
    2. Don't enumerate the KCs or steps within the list.

--- Example format below ---
{0: {"list_KCs": [<KC 1>, ..., <KC M0>], "question": <question_text>, "
    list_sol_steps":[<step 1>, ..., <step N0>]},
 1: {"list_KCs": [<KC 1>, ..., <KC M1>], "question": <question_text>, "
    list_sol_steps":[<step 1>, ..., <step N1>]},
 2: {"list_KCs": [<KC 1>, ..., <KC M2>], "question": <question_text>, "
    list_sol_steps":[<step 1>, ..., <step N2>]}}
```

**User Prompt Template**

```
Question: <QUESTION TEXT>
Solution steps: <SOLUTION STEPS>
Knowledge concepts: <LIST OF KCs>
```

## H.1 Example Artificial Questions

We present one example original question from the dataset along with its solution steps and knowledge concepts (KCs). Then, we show two artificial questions generated based on it using the prompt described in this section. New conceptual elements or solution changes introduced by the model are highlighted in purple.

**Original Question**

- **Question:** Eddie went to the stationery store to buy stationery for the teacher. The unit prices of the stationery are as follows: a mechanical pencil is 4 yuan, a pencil box is 26 yuan, a pen is 45 yuan, and a school bag is 128 yuan. Eddie wants to buy 32 pencils, 24 pencil cases, 65 pens, and 9 school bags. How much does Eddie want to spend in total?

- **Solution Steps:**
    1. Calculate the cost of 32 mechanical pencils: 4 yuan each x 32 = 128 yuan.
    2. Calculate the cost of 24 pencil boxes: 26 yuan each x 24 = 624 yuan.
    3. Calculate the cost of 65 pens: 45 yuan each x 65 = 2925 yuan.
    4. Calculate the cost of 9 school bags: 128 yuan each x 9 = 1152 yuan.
    5. Add all the costs together: 128 + 624 + 2925 + 1152 = 4829 yuan.

- **KCs: a)** Understanding multiplication, **b)** Solving multiplication and division problems in real-world contexts, **c)** Adding and subtracting multi-digit whole numbers

**Artificial Generation 1**

- **Question:** Eddie is buying supplies for a school event. The prices are as follows: a box of markers costs 4 yuan, a pack of paper costs 12 yuan, a set of paints costs 30 yuan, and a roll of tape costs 8 yuan. Eddie wants to buy 50 boxes of markers, 40 packs of paper, 20 sets of paints, and 15 rolls of tape. How much will Eddie spend in dollars, if 1 dollar equals 7 yuan?

- **Solution Steps:**
    1. Calculate the cost of 50 boxes of markers: 4 yuan each x 50 = 200 yuan.
    2. Calculate the cost of 40 packs of paper: 12 yuan each x 40 = 480 yuan.
    3. Calculate the cost of 20 sets of paints: 30 yuan each x 20 = 600 yuan.
    4. Calculate the cost of 15 rolls of tape: 8 yuan each x 15 = 120 yuan.
    5. Add all the costs together: 200 + 480 + 600 + 120 = 1400 yuan.
    6. Convert the total cost to dollars: 1400 yuan / 7 = 200 dollars.

- **KCs: a)** Understanding multiplication, **b)** Solving multiplication and division problems in real-world contexts, **c)** Adding and subtracting multi-digit whole numbers, **d)** Understanding unit conversion

**Artificial Generation 2**

- **Question:** Alice is purchasing items for a charity event. The prices are as follows: a notebook costs 6 yuan, a folder costs 10 yuan, a calculator costs 60 yuan, and a backpack costs 150 yuan. Eddie wants to buy 20 notebooks, 30 folders, 10 calculators, and 5 backpacks. If Alice receives a 10% discount on the total purchase, how much will he spend?

- **Solution Steps:**
    1. Calculate the cost of 20 notebooks: 6 yuan each x 20 = 120 yuan.
    2. Calculate the cost of 30 folders: 10 yuan each x 30 = 300 yuan.
    3. Calculate the cost of 10 calculators: 60 yuan each x 10 = 600 yuan.

4. Calculate the cost of 5 backpacks: 150 yuan each x 5 = 750 yuan.
5. Add all the costs together: 120 + 300 + 600 + 750 = 1770 yuan.
6. Calculate the discount: 10% of 1770 yuan = 177 yuan.
7. Subtract the discount from the total cost: 1770 - 177 = 1593 yuan.

- **KCs:** **a)** Understanding multiplication, **b)** Solving multiplication and division problems in real-world contexts, **c)** Adding and subtracting multi-digit whole numbers, **d)** Understanding percentages

# I   Visualization of Conceptual Growth (Extended)

Figure 9 demonstrates how the knowledge state of a student evolves for a wide range of policies and also non-RL baselines. Overall, the results are consistent with our earlier findings.

The non-RL baselines are not targeting the weaknesses of the students, and in fact, they even cause a decline in their knowledge state. On the other hand, most RL policies provide meaningful knowledge gains across multiple concepts. Further, we observe better improvements for the RL policies improved by our model based value estimation (w/ MVE). When tested on an extended corpus, the default version of the continuous action models perform poorly. Yet, our MVE approach provides large gains to these models on the extended corpus. This demonstrates that these value-based continuous action models can adapt to new set of questions with the ability of planning ahead via our MVE.

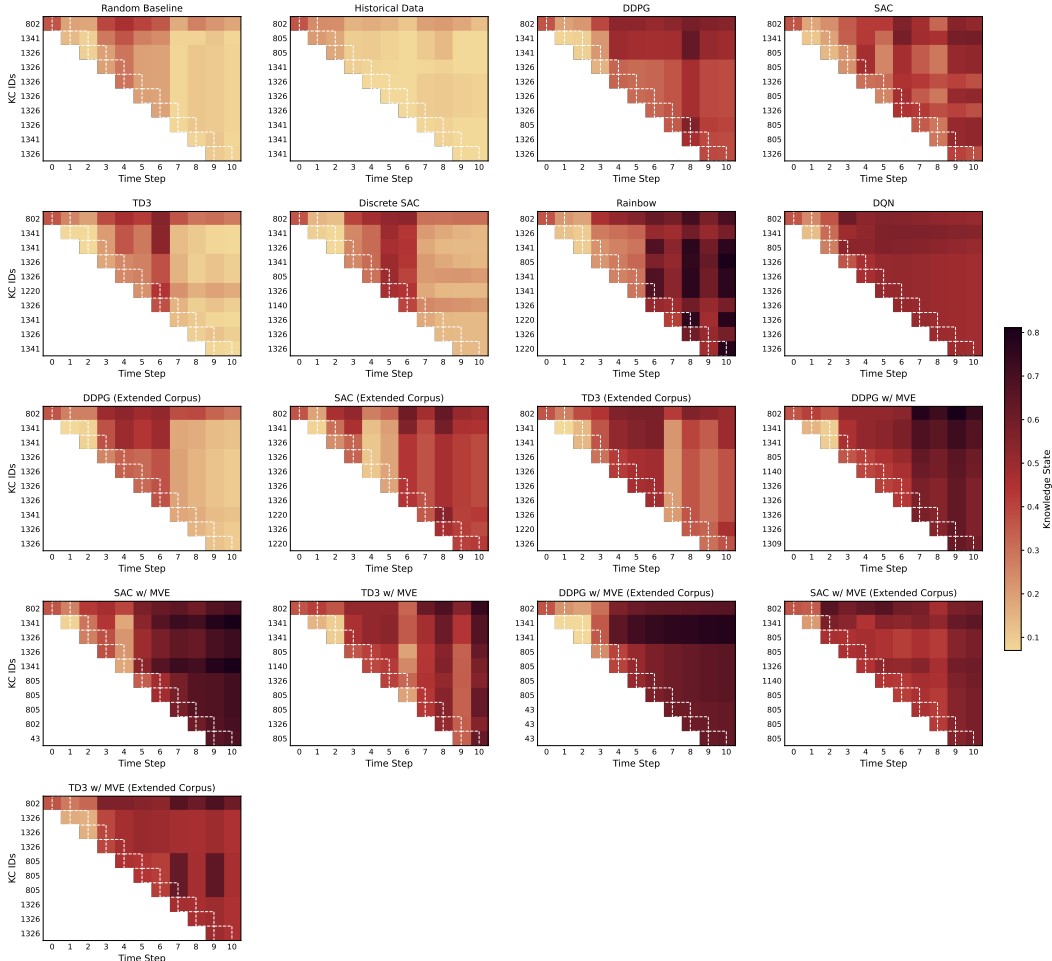

Figure 9: Knowledge state evolution for a single student across various policies and non-RL baselines (Task 4). At each step, the weakest KC is targeted, and its knowledge trajectory is shown. White dashed boxes mark changes after each recommendation. A KC may appear more than once if it remains the weakest. KC IDs are for visualization only; our framework operates on semantic KC embeddings, not IDs

# J Generalization to an Additional Dataset

To evaluate the generalizability of our framework, we conduct further experiments on the Eedi dataset [25]. This large-scale dataset contains 2,324,162 learning interactions from 47,560 students. The corpus consists of 4,019 unique questions which are annotated with 1,215 distinct knowledge concepts (KCs).

Table 2: Global Knowledge Improvement on the Eedi Dataset (Reported: % Max. Improvement)

| Random Policy | Historical Data | DDPG | DDPG w/ MVE | SAC | SAC w/ MVE | TD3 | TD3 w/ MVE | PPO | TRPO | Discrete SAC | C51 | Rainbow | DQN |
|---|---|---|---|---|---|---|---|---|---|---|---|---|---|
| -2.31 | -0.21 | 16.65 | 18.87 | 8.97 | 20.08 | 9.62 | 16.85 | 1.78 | 2.83 | 18.81 | 20.83 | 37.00 | 30.51 |

Specifically, we evaluate our framework on the Eedi dataset using the Global Knowledge Improvement task (Task 1) from the main paper, where the objective is to maximize the average knowledge improvement across all KCs. The results, presented in Table 2, confirm the key findings from our primary experiments on the XES3G5M dataset. Notably, our model-based value estimation (MVE) again provides a significant performance boost to continuous value-based RL methods, and discrete-action algorithms also perform strongly, reinforcing the robustness of our approach.

It is important to note that the reported percentage improvements on the Eedi dataset appear lower than those on XES3G5M. This is because the maximum possible absolute improvement is substantially larger for Eedi (0.56 vs. 0.21). As a result, students must achieve larger absolute knowledge gains to reach full mastery. In absolute terms, the improvements on Eedi are comparable to those on XES3G5M, even though the percentages appear smaller.

# K  Future Direction: Incorporating Misconception Analysis

An important future direction for enhancing personalized learning is the analysis of students' incorrect responses to a given question (e.g., choosing distractor B vs. C). Different incorrect choices often represent distinct cognitive models and specific knowledge confusions. This type of error-specific analysis could reveal not just what students got wrong, but why they made specific errors. This section outlines current data limitations that hinder this approach and provides a detailed recipe for how the ExRec framework could be extended to incorporate such a misconception analysis.

**Data Limitation.** A practical challenge for this approach is that most existing knowledge tracing datasets, including those used in our work, only provide a binary correctness label for each student interaction. The specific incorrect option chosen by the student is not recorded, making it impossible to distinguish between different error patterns or map them to specific misconceptions.

Should future datasets become available that include students' specific answer choices, the `ExRec` framework could be naturally extended to model misconceptions. We outline a detailed recipe below:

1. **Module 1 (LLM Annotation)** could be augmented with the following steps to produce fine-grained labels that inform the representation learning in Module 2. This would enable the model to capture not only *what* misconception occurred, but also *where* it occurred in the reasoning process:

    (a) **Misconception annotation:** Given the question, its solution steps, KCs, and a specific incorrect option, the LLM would be prompted to annotate the underlying misconception reflected by that choice.

    (b) **Incorrect solution generation:** Using the correct solution steps, the identified misconception, and the incorrect option, the LLM would then generate an incorrect solution that deviates from the correct one at the point where the misconception is applied.

    (c) **Incorrect step–misconception mapping:** Finally, the LLM would pinpoint exactly which step(s) in the generated incorrect solution correspond to the annotated misconception.

2. **Module 2 (Contrastive Learning)** could then incorporate these new annotations to learn richer representations:

    (a) **New token:** A new special token, `[MC]`, would be introduced to mark misconception text, alongside the existing tokens (`[Q]`, `[S]`, `[KC]`), allowing the encoder to learn distinct representations.

    (b) **Extended loss function:** In addition to the original contrastive loss $L_s$ for correct solution steps, a new loss term $L_{mc}$ would be introduced to align the embeddings of incorrect solution steps with their corresponding misconception embeddings.

3. **Module 3 (Knowledge Tracing)** would be adapted to process these new, richer inputs:

    (a) **Input embedding:** Instead of combining the question embedding with a simple binary correctness indicator, the input would be formed using either the correct solution step embeddings (if the student answered correctly) or the incorrect solution step embeddings (if the student's answer is linked to a known misconception).

    (b) **Dual-task prediction:** The model would have two outputs. The primary output for performance prediction would remain. A second output would predict the misconception embedding, with an auxiliary loss based on the cosine similarity to the ground-truth misconception (this loss would be masked for correct answers). This avoids a rigid categorical formulation and allows for the seamless integration of new misconceptions.

    (c) **KC calibration:** The calibration process for knowledge states would remain unchanged.

4. **Module 4 (Reinforcement Learning)** would require no structural changes. The KT model, now enhanced with misconception modeling, would serve as a more sophisticated RL environment, inherently providing the necessary state transitions and rewards to the RL agent.

We believe this extension represents a powerful direction for future research. The proposed recipe demonstrates both the feasibility and the natural fit of misconception analysis within the ExRec framework. While current dataset limitations preclude its immediate implementation, we hope this detailed outline inspires future work in data collection and modeling to better understand the nuances of student learning.

