# OpenReview forum: "Personalized Exercise Recommendation with Semantically-Grounded Knowledge Tracing"
_NeurIPS.cc/2025/Conference — NeurIPS 2025 poster_

### Official Review · Reviewer_SGqL · 2025-06-30

**Clarity:** 4
**Significance:** 3
**Originality:** 3
**Rating:** 5
**Confidence:** 5

**Summary:**

This paper presents a knowledge tracing (KT) algorithm based on reinforcement learning (RL) for personalized education.

**Questions:**

Do the authors have any plan how to ensure QA of the generated solutions by LLMs?

**Ethical Concerns:**

["NO or VERY MINOR ethics concerns only"]

**Final Justification:**

I have read the authors' rebuttal. The authors have addressed by concern and also added additional experimental results.

This work has promise in personalized learning. I have raised my score.

**Limitations:**

The authors have used only one dataset.
They can also explore relevant K-12 math dataset on their proposed model, such as the ASSISTments datasets.

**Quality:**

3

**Strengths And Weaknesses:**

Strength
1.	Produces interpretable learning trajectories via reinforcement learning (RL)

2.	KT generalizes to new, unseen question (Section 6.1)

3.	Experimented on four evaluation tasks: global knowledge improvement, knowledge improvement in practiced knowledge component (KC), weakest KC, upcoming KC.

4.	Compact representation of states, that do not need to process the student’s entire learning sequence

Weakness
1.	The limitations specified in the introduction “they often rely on ID-based embeddings, which neglects the semantics of question”—is not correct. The first DKT paper was on question ID-based embeddings. But after that, the weakness has been addressed by the community. Most DKT algorithms rely of embeddings of KC, such as AKT [1].
Embeddings of questions has been explored by researchers’ way before the LLM era, see a 2018 paper: EERNN[2], a 2020 paper: RAKT[3], a 2022 paper: SRL-KT [4].

2.	The KC representation in Section 4.1 is a problematic, in my opinion. Prior deep learning era, KC annotation was performed by domain experts with multiple rounds of review.
The proposed approach annotated KC by a sampling method through LLM with no expert intervention. The step can be performed by a machine-teaching or human-in-the loop fashion.
Similarly, the solution step generation is done by LLM, but no quality assurance is performed.

References

1.	Ghosh, A., Heffernan, N. and Lan, A.S., 2020, August. Context-aware attentive knowledge tracing. In Proceedings of the 26th ACM SIGKDD international conference on knowledge discovery & data mining (pp. 2330-2339).

2.	. Su, Y., Liu, Q., Liu, Q., Huang, Z., Yin, Y., Chen, E., Ding, C., Wei, S. and Hu, G., 2018, April. Exercise-enhanced sequential modeling for student performance prediction. In Proceedings of the AAAI conference on artificial intelligence (Vol. 32, No. 1).

3.	 Pandey, S. and Srivastava, J., 2020, October. RKT: relation-aware self-attention for knowledge tracing. In Proceedings of the 29th ACM international conference on information & knowledge management (pp. 1205-1214).

4.	Farhana, E., Rutherford, T. and Lynch, C.F., 2022, June. Predictive student modelling in an online reading platform. In Proceedings of the AAAI conference on artificial intelligence (Vol. 36, No. 11, pp. 12735-12743).

---

> ### Author Rebuttal · Authors · 2025-07-30
>
> Thank you very much for your thoughtful review and for recognizing the potential of our work!
>
> Your decision to support this submission gives us the opportunity to bring a foundational contribution to the community. **We strongly believe that ExRec opens up a scalable and modular pathway for future research in personalized education**, enabling researchers to plug in their own datasets, KT models, or RL policies and push the boundaries of exercise recommendation systems. We truly appreciate your feedback in helping us refine and strengthen this vision.
>
> Below, please find our clarifications to the points you raised.
>
> **W1. The limitations specified in the introduction “they often rely on ID-based embeddings, which neglects the semantics of question”—is not correct. The first DKT paper was on question ID-based embeddings. But after that, the weakness has been addressed by the community. Most DKT algorithms rely of embeddings of KC, such as AKT [1]. Embeddings of questions has been explored by researchers’ way before the LLM era, see a 2018 paper: EERNN[2], a 2020 paper: RAKT[3], a 2022 paper: SRL-KT [4].**
>
> Thank you very much for your thoughtful comment! We agree that there has been meaningful progress in incorporating question and KC semantics into KT models, and we appreciate your references to EERNN, RAKT, and SRL-KT. However, we respectfully believe that the broader problem of relying on ID-based embeddings is **far from solved**, especially when viewed in the context of **exercise recommendation** or more generalizable KT frameworks.
>
> Indeed, our framework differs from prior approaches not just in using semantic information, but in **how** we use it. The cited models typically rely on shallow integration, such as pre-trained word embeddings or static text encoders. In contrast, **we train our question representations end-to-end via a contrastive learning module that explicitly aligns questions (and their solution steps) with corresponding knowledge concepts.** This yields richer, more interpretable embeddings that better support both generalization and downstream tasks like RL-based recommendation.
>
> Further, even the most **recent and advanced KT models** [e.g., 1,2,3,4,5,6] still primarily rely on ID-based embeddings or even only KC embeddings. This highlights that, while promising work exists, the field has not fully moved beyond this limitation. We appreciate the opportunity to clarify this distinction and will revise our framing accordingly.
>
> [1] Li, Qing, Zhijun Huang, Jianwen Sun, Xin Yuan, Shengyingjie Liu, and Zhonghua Yan. "HKT: Hierarchical structure-based knowledge tracing." Information Processing & Management. 2025.
>
> [2] Cheng, Weihua, Hanwen Du, Chunxiao Li, Ersheng Ni, Liangdi Tan, Tianqi Xu, and Yongxin Ni. "Uncertainty-aware Knowledge Tracing." AAAI. 2025.
>
> [3] Bai, Youheng, Xueyi Li, Zitao Liu, Yaying Huang, Mi Tian, and Weiqi Luo. "Rethinking and improving student learning and forgetting processes for attention based knowledge tracing models." AAAI. 2025.
>
> [4] Hou, Mingliang, Xueyi Li, Teng Guo, Zitao Liu, Mi Tian, Renqiang Luo, and Weiqi Luo. "Cognitive Fluctuations Enhanced Attention Network for Knowledge Tracing." AAAI. 2025.
>
> [5] Bai, Youheng, Xueyi Li, Zitao Liu, Yaying Huang, Teng Guo, Mingliang Hou, Feng Xia, and Weiqi Luo. "csKT: Addressing cold-start problem in knowledge tracing via kernel bias and cone attention." Expert Systems with Applications. 2025.
>
> [6] Shen, Xiaoxuan, Fenghua Yu, Yaqi Liu, Ruxia Liang, Qian Wan, Kai Yang, and Jianwen Sun. "Revisiting knowledge tracing: A simple and powerful model." ACM International Conference on Multimedia. 2024.
>
>
> **W2: The KC annotation in Section 4.1 is a problematic, in my opinion. Prior deep learning era, KC annotation was performed by domain experts with multiple rounds of review. The proposed approach annotated KC by a sampling method through LLM with no expert intervention. The step can be performed by a machine-teaching or human-in-the loop fashion. Similarly, the solution step generation is done by LLM, but no quality assurance is performed.**
>
> Thank you for raising this important point! We respectfully believe that the absence of expert intervention in our KC annotation process is not a drawback, but a key strength. Our aim is to reduce reliance on time-consuming and potentially inconsistent expert annotations to enable scalable and adaptable pipelines for new datasets and domains. That said, our framework is modular by design: teams with access to high-quality expert-labeled KCs can bypass this step and proceed directly to the subsequent modules.
>
> Regarding quality assurance, we build upon recent studies [1,2] that rigorously evaluated the use of LLMs for KC annotation. These works conducted both automated and human evaluations, comparing LLM-generated annotations with those provided in standard KT datasets. In both cases, the LLM-based annotations were consistently preferred, particularly for their clarity, generalizability, and semantic coherence: they often outperformed the dataset-provided expert labels.
>
> Finally, we agree that quality control remains important, and our approach naturally accommodates optional human review or refinement where needed. Our contribution lies in demonstrating that high-quality KC annotations can be generated automatically and reliably. This way, we significantly expand the accessibility and applicability of personalized educational systems.
>
>
> **Q1: Do the authors have any plan how to ensure QA of the generated solutions by LLMs?**
>
> Thank you very much for raising this important point! While enhancing the mathematical reasoning abilities of LLMs is beyond the scope of our work, we fully agree that **ensuring the quality of generated solution steps is essential** for maintaining the validity of our framework.
>
> To this end, we leveraged the fact that the XES3G5M dataset includes ground-truth answers. After generating solution steps via the LLM, we conducted a second round of LLM-based evaluation, comparing the final computed answer within the generated steps against the correct answer. Since extracting the answer from free-form output is non-trivial, this LLM-assisted comparison proved both **scalable and reliable**. We observed a correctness rate of over 98%, with most discrepancies involving minor rounding issues.
>
> Crucially, in our framework, what truly matters is **not the final numeric accuracy**, but the **verbal description of the strategy** expressed in the solution steps. This is because our contrastive learning and semantic embedding modules compute representations based on the **textual and conceptual content** of entire sentences, where numerical tokens contribute minimally to the embedding. Therefore, as long as the reasoning is sound and well-articulated, minor arithmetic deviations do not undermine the learning signal.
>
> Still, for flagged errors, we prompted the LLM again (this time providing the correct answer) to encourage better alignment. We further validated the QA process by manually reviewing 100 random samples marked as correct by the evaluator, and confirmed that they were indeed accurate.
>
> While not central to our core contribution, we will describe this procedure in detail in the implementation section. We appreciate your suggestion to include it!
>
>
> **L1:The authors have used only one dataset. They can also explore relevant K-12 math dataset on their proposed model, such as the ASSISTments datasets.**
>
> Thank you for the suggestion. We initially considered using ASSISTments, which exists in several versions across different years. However, in the versions accessible to us, the question content was either **completely unavailable** or embedded in **highly noisy HTML**, often rendering the items **unreadable even for human annotators.** Given that our framework critically relies on understanding the semantic content of questions for KC annotation and exercise representation, ASSISTments was unfortunately not a viable option for a meaningful evaluation.
>
> That said, we took your suggestion seriously and extended our evaluation to **another large-scale online math learning dataset, EEDI**, which offers clean and verified textual content. This dataset includes 47,560 students, 4,019 unique questions, and a total of 2,324,162 interactions. It shares many real-world characteristics with the XES3G5M dataset and is provided by the same organization behind the NIPS34 benchmark. Its questions were OCR-transcribed and reviewed by EEDI’s internal team, making it well-suited for semantic analysis.
>
> **Task 1: Global Knowledge Improvement. (Reported: % Max. Improvement as in paper)**
>
> |Random Policy|Historical Data|DDPG|DDPG w/ MVE|SAC|SAC w/ MVE|TD3|TD3 w/ MVE|TRPO|PPO|Discrete SAC|C51|Rainbow|DQN|
> |--|--|--|--|--|--|--|--|--|--|--|--|--|--|
> |-2.31|-0.21|16.65|18.87|8.97|20.08|9.62|16.85|1.78|2.83|18.81|20.83|37.00|30.51|
>
> *Note:* The reported improvements on the EEDI dataset appear lower than those on XES3G5M because the maximum possible absolute improvement is much larger (0.56 vs. 0.21). As a result, students must achieve larger absolute knowledge gains to reach full mastery. In absolute terms, the improvements on EEDI are comparable to those on XES3G5M, even though the percentages appear smaller.
>
> Thanks to the flexibility of our framework, we were able to seamlessly apply our entire pipeline to EEDI, and we report the results in the revised version.

---

> > ### Comment · Area_Chair_jyvD · 2025-08-05
> >
> > The authors have provided a rebuttal to your comments, and it's an important part of the review process to give their response careful consideration. Please take a moment to review their rebuttal and provide any follow-up comments. This will help ensure there’s sufficient time for discussion and any necessary follow-up.
> >
> > Best regards,
> >
> > AC

---

> > ### Comment · Reviewer_SGqL · 2025-08-07
> >
> > I have read your rebuttal and raised your score, as all of my concerns have been addressed.

---

> > > ### Author Response · Authors · 2025-08-07
> > >
> > > We sincerely thank the reviewer for their thoughtful engagement and for re-evaluating our work!
> > >
> > > We truly appreciate your recognition, and we're grateful that our clarifications addressed your concerns.
> > >
> > > Thank you again for your support in helping bring this work to the broader research community!

---

### Official Review · Reviewer_ABnB · 2025-07-01

**Clarity:** 3
**Significance:** 3
**Originality:** 2
**Rating:** 5
**Confidence:** 4

**Summary:**

This paper presents a knowledge tracing (KT) based framework for personalized exercise recommenddation. The proposal annotates the knowledge concepts of questions and learning their semantic representations to training KT models, while optimizing various reinforcement learning methods. Additionally, a model-based value estimation approach is designed to improve the standard Q-learning-based continuous reinforcement learning methods. Experiments on one publicly available knowledge tracing datasets are conducted to validate the effectiveness of the proposal.

**Questions:**

1. Missing several related works:
The literature review lacks several important recent works on knowledge tracing based exercise recommendation methods, such as
“Exercise recommendation method based on knowledge tracing and concept prerequisite relations”, CCF Transactions on Pervasive Computing and Interaction, 2022;
“Exercise recommendation based on knowledge concept prediction”, Knowledge-Based Systems, 2020;
“Pedagogical interventions in SPOCs: Learning behavior dashboards and knowledge tracing support exercise recommendation”, IEEE Transactions on Learning Technologies, 2023;
“Enhanced personalized learning exercise question recommendation model based on knowledge tracing”, International Journal of Advances in Intelligent Informatics, 2024;
“MLKT4Rec: Enhancing Exercise Recommendation Through Multitask Learning With Knowledge Tracing”, IEEE Transactions on Computational Social Systems, 2024;
“Knowledge modeling via contextualized representations for LSTM-based personalized exercise recommendation”, Information Sciences, 2020;

The authors should provide a comprehensive discussion of these works and explicitly state how their proposal advances beyond existing approaches. Furthermore, empirical comparisons with these methods would better demonstrate the effectiveness of the proposal.

2. Limited dataset evaluation:
The proposal is validated on only one knowledge tracing dataset. Several well-established benchmark datasets are available, such as NIPS34 and EdNet. The authors are encouraged to add these representative datasets to provide more comprehensive validation of their proposal.

3. Lack of mutual impact analysis:
While the proposal is designed for exercise recommendation, it would be valuable to investigate whether the proposal could, in turn, improve knowledge tracing performance.

4. Insufficient efficiency analysis:
The authors are encouraged to report the inference time performance, as it directly impacts the feasibility of real-time implementation in educational systems.

**Ethical Concerns:**

["NO or VERY MINOR ethics concerns only"]

**Final Justification:**

Through discussions with the authors, all of my concerns—including (i) Missing several related works, (ii) Limited dataset evaluation, (iii)  Lack of mutual impact analysis and (iv) Insufficient efficiency analysis—have been satisfactorily addressed.

I believe this paper makes a promising contribution to the field of educational data mining, and I am therefore increasing my overall score.

**Limitations:**

Yes

**Quality:**

3

**Strengths And Weaknesses:**

Strengths
1. This paper is well-structured, and a decent amount of technical details are given.
2. The idea of modelling personalized exercise recommendation with the semantically-grounded knowledge tracing is interesting.

Weaknesses
1. Missing several related works
2. Limited dataset evaluation
3. Lack of mutual impact analysis
4. Insufficient efficiency analysis

---

> ### Author Rebuttal · Authors · 2025-07-30
>
> Thank you for your thoughtful and constructive review! We appreciate your recognition of the novelty and potential of our work.
>
> Our goal with ExRec is to provide a **foundational, extensible framework** that bridges knowledge tracing and reinforcement learning for personalized education. Built for **flexibility, modularity, and scalability**, ExRec allows researchers to plug in their own datasets, use advanced KT models, and test alternative RL strategies in a unified, semantically grounded environment.
>
> We believe NeurIPS is the ideal venue to **catalyze the next wave of research in personalized education**, and we have addressed all your suggestions to further strengthen and clarify our contributions.
>
> **Q1: Missing several related works**
>
> Thank you for your valuable suggestions. Our original submission focused on works aiming to improve knowledge state using RL-based frameworks built on simulated student behavior, which aligns with our central goal: learning a recommendation policy that optimizes long-term knowledge improvement through semantically grounded KT.
>
> Following your suggestion, **we now broaden our discussion to include a wider range of exercise recommendation approaches.** While relevant, these works do **not directly optimize knowledge state**, nor do they develop a trainable RL-based policy. Most rely on rule-based heuristics, optimize simpler proxy metrics (e.g., difficulty, novelty), and generally overlook semantic representations of questions and KCs.
>
> We elaborate on each paper below in the order provided.
>
> `Exercise recommendation method based on knowledge tracing and concept prerequisite relations:` This method recommends KCs rather than actual exercises, treating all exercises within a KC as interchangeable. It relies on manually defined prerequisite graphs and uses only KC IDs, ignoring the semantic content of both KCs and questions. The recommendation is rule-based and static. It *neither* learns from data *nor* simulates how student knowledge evolves over time.
>
> `Exercise recommendation based on knowledge concept prediction:` This work models KC mastery and recommends KCs based on predicted gains. The recommendation logic is rule-based and *not* trainable. It does *not* model question semantics, nor does it use RL or simulate student behavior. It focuses on proxy objectives like difficulty or novelty, whereas ExRec directly optimizes for actual knowledge state improvement.
>
> `Pedagogical interventions in SPOCs: Learning behavior dashboards and KT support:` This work applies KT to estimate students’ knowledge but does not perform exercise recommendation. Instead, it provides a dashboard interface for instructors to decide what to recommend. The KT model relies on *hand-crafted* features, unlike ExRec, which operates directly on question semantics and automates the full pipeline.
>
> `Enhanced personalized learning exercise question recommendation based on KT:` This approach predicts question difficulty and filters exercises accordingly, based on instructor-specified difficulty ranges. It applies KT but does *not* simulate student learning or use RL. It assumes external input of difficulty to guide recommendation and does not attempt to optimize knowledge improvement directly.
>
> `MLKT4Rec: Enhancing Exercise Recommendation Through Multitask Learning with KT:` MLKT4Rec casts recommendation as a multitask supervised problem, jointly predicting correctness and recommending exercises to maximize immediate success. Therefore, it does *not* prioritize long-term knowledge gain. It does *not* incorporate RL, simulate knowledge gain, or model semantic content or KCs. In contrast, ExRec optimizes for long-term knowledge improvement, incorporates LLM-based KC annotations, and enables generalization to unseen questions.
>
> `Knowledge modeling via contextualized representations for LSTM-based personalized exercise recommendation:` This method uses LSTM and attention to recommend exercises based on historical interactions. It focuses on predicting immediate performance without RL, semantic modeling, or KC representations. ExRec advances this line by using traced knowledge states, semantically enriched representations, and RL to guide students toward lasting knowledge gains.
>
> **->** We will incorporate these works into our extended Related Work section to clearly position ExRec in relation to these approaches and to highlight our unique contributions.
>
> **Q2: Limited dataset evaluation: The proposal is validated on only one knowledge tracing dataset. Several well-established benchmark datasets are available, such as NIPS34 and EdNet. The authors are encouraged to add these representative datasets to provide more comprehensive validation of their proposal.**
>
> Thank you for the suggestion. While we agree that broader dataset evaluation is important, our framework specifically leverages the semantic content of questions, which is central to our contribution. Unfortunately, the suggested datasets are incompatible with this requirement:
>
> - **EdNet** does not provide question text — only question IDs — making it unusable for semantically grounded recommendation.
>
> - **NIPS34** includes only *image*-based questions, many with geometric content. Converting them to meaningful text requires reliable multi-modal understanding, which is beyond this paper’s scope. However, we appreciate this suggestion and will include it as a limitation and a future direction.
>
> To address your feedback, **we extended our evaluation to the EEDI dataset**, developed by the same creators as NIPS34. EEDI provides OCR-transcribed and manually verified textual question content. Therefore, it is well-suited for our framework. It also reflects real-world conditions, with 47,560 students, 4,019 unique questions, and over 2.3M interactions.
>
> Thanks to ExRec’s modular design, we seamlessly applied our full pipeline to this dataset. Please find the results below:
>
> **Task 1: Global Knowledge Improvement.** (Reported: % Max. Improvement as in paper.)
>
> |Random Policy|Historical Data|DDPG|DDPG w/ MVE|SAC|SAC w/ MVE|TD3|TD3 w/ MVE|PPO|TRPO|Discrete SAC|C51|Rainbow|DQN|
> |--|--|--|--|--|--|--|--|--|--|--|--|--|--|
> |-2.31|-0.21|16.65|18.87|8.97|20.08|9.62|16.85|1.78|2.83|18.81|20.83|37.00|30.51|
>
> *Note:* The reported improvements on the EEDI dataset appear lower than those on XES3G5M because the maximum possible absolute improvement is much larger (0.56 vs. 0.21). As a result, students must achieve larger absolute knowledge gains to reach full mastery. In absolute terms, the improvements on EEDI are comparable to those on XES3G5M, even though the percentages appear smaller.
>
> We are grateful for the opportunity to demonstrate that **our framework generalizes easily to other large-scale math datasets**. This has been a great opportunity to demonstrate its practical applicability and extensibility.
>
>
> **Q3: Lack of mutual impact analysis: While the proposal is designed for exercise recommendation, it would be valuable to investigate whether the proposal could, in turn, improve knowledge tracing performance.**
>
> Thank you very much for your valuable (and quite interesting) suggestion! As a disclaimer, please correct us if we misinterpreted your suggestion — if that’s the case, we are more than happy to investigate it more deeply!
>
> Following your suggestion, we treated the generated trajectories through our RL environment as the synthetic dataset (i.e., the new question-answer pairs). We then appended this synthetic dataset to the original KT dataset and performed another round of KT training.
>
> After the above procedure, we found that the AUROC score of KT model dropped from 81.65 to 81.32. We believe the main reason is that the original dataset was large enough to train our KT model and that the synthetic dataset propagated the errors of a KT model that caused a decrease in the performance.
>
> To confirm the above hypothesis, we trained another KT model (from scratch) with only 10% of the existing students in the dataset and repeated the above procedure of producing synthetic data through our RL environment. This time, we observed an increase in the prediction performance from 79.41 to 79.98 AUROC.
>
> These findings suggest that our framework can serve not only as a recommendation policy but also as a data augmentation tool to improve KT performance. This is particularly relevant in **low-data regimes.**
>
>
> **->** We will incorporate this analysis into the paper as an additional use case, especially highlighting its value when the number of available student interactions is limited.
>
> **Q4: Insufficient efficiency analysis: The authors are encouraged to report the inference time performance, as it directly impacts the feasibility of real-time implementation in educational systems.**
>
> Thank you very much for this insightful suggestion! While we had already reported the training time and hardware requirements of our computationally intensive modules in Appendix B, we agree that **inference time** is an important factor for real-time deployment in educational systems.
>
> In response, we conducted additional experiments to measure inference efficiency. Below, we report:
>
> - **KT training** takes around 2.5 hours on a single NVIDIA Quadro RTX 6000 GPU with 32 GB of memory.
>
> - **KT inference per question** takes  8.7 milliseconds on a single NVIDIA Quadro RTX 6000 GPU with 24 GM of memory.
>
> - **RL inference per exercise recommendation** takes 78.5 milliseconds on a single NVIDIA Quadro RTX 6000 GPU with 24 GM of memory.
>
> We will add these details to our Appendix B.

---

> > ### Comment · Area_Chair_jyvD · 2025-08-05
> >
> > The authors have provided a rebuttal to your comments, and it's an important part of the review process to give their response careful consideration. Please take a moment to review their rebuttal and provide any follow-up comments. This will help ensure there’s sufficient time for discussion and any necessary follow-up.
> >
> > Best regards,
> >
> > AC

---

> > ### Comment · Reviewer_ABnB · 2025-08-06
> >
> > Thank you for the rebuttal. Although many of my earlier concerns have been addressed, two fundamental issues remain.
> >
> > 1\. **Lack of Empirical Benchmarks**: The manuscript now surveys the relevant literature, yet it still lacks direct empirical comparisons with those methods. Without such benchmarks, the claimed superiority of the proposed approach remains unsubstantiated.
> >
> > 2\. **Insufficient Evidence for the Data Augmentation Claim**: The authors argue that their framework can serve as a "data augmentation tool... particularly in low-data regimes." This is an interesting claim, yet the supporting evidence remains insufficient. To support this claim, the following analyses may be considered:
> >
> > * **Investigation of Synthetic-Data Properties.** The "error propagation" hypothesis is plausible but not sufficiently proven. A deeper analysis of the synthetic data itself (e.g., its quality, diversity, and distributional similarity to the real data) is needed to better understand why it helps in low-data regimes but hurts in high-data regimes. An alternative hypothesis, such as a regularization effect, may also be considered.
> >
> > * **Comparative Analysis with Baseline Augmentation Methods.** To validate the claim of being a useful "data augmentation tool,"  it should be benchmarked against simpler, standard techniques (e.g., straightforward oversampling).
> >
> > * **Systematic Ablation Study to Define the “Low-Data Regime”.** The conclusion is based on a single "low-data" point (10%). An ablation study on varying data sizes (e.g., 20%, 50%) would more clearly define the conditions under which the proposal is beneficial.

---

> > > ### Author Response · Authors · 2025-08-07
> > >
> > > We sincerely thank the reviewer for their continued engagement and careful reading of our rebuttal. We respectfully respond to the two remaining concerns below:
> > >
> > > ---
> > >
> > > ### **1. Lack of Empirical Benchmarks**
> > >
> > > We believe this concern stems from a misunderstanding of the fundamental objectives of our framework versus the cited works. We summarize our perspective below:
> > >
> > > - **As an important observation**: *Although these works are now collectively deemed as essential baselines, most of them do not cite or compare against each other*.
> > >   - This reflects the fragmented nature of the current literature.
> > >   - In contrast, our work not only situates itself more broadly, such as integrating knowledge tracing, semantic representations, and reinforcement learning, but also lays the groundwork for more unified and rigorous evaluation going forward.
> > >
> > > - **Our core novelty lies in optimizing *long-term knowledge improvement* through simulated student behavior and reinforcement learning (RL)**.
> > >   In contrast, *none* of the cited works simulate student knowledge dynamics, leverage RL, or consider the semantic representations of questions and knowledge concepts (KCs).
> > >
> > > - **Most cited works aim to *predict the next question* a student would attempt, not to improve knowledge state.**
> > >   - Their evaluation metrics (e.g., Hit Rate@K, NDCG, MRR, Accuracy, Precision) all measure predictive performance against ground-truth next interactions.
> > >   - However, *what a student chooses to attempt next is not necessarily optimal for learning*: it may reflect prior habits, biases, or unstructured curricula.
> > >
> > > - ⚠️ **We kindly ask the reviewer to note that we *already compare ExRec against this predictive baseline*** in all our experiments:
> > >   - Our *Historical Data* baseline simulates the behavior of a perfect next-step predictor.
> > >   - If these methods were ideal at their task (i.e., perfectly predict the next question attempted), this baseline would reflect their upper bound.
> > >   - Yet, as shown in all tables, *this does not translate to meaningful knowledge improvement*.
> > >
> > > - **This validates our claim: fitting historical data is not sufficient: simulated student behavior and RL are necessary.**
> > >
> > > - **These methods are fundamentally incomparable to ExRec**, as their objectives do not align with ours. Even if one insisted on a head-to-head comparison:
> > >   - *None of the cited works have released public code*.
> > >   - Given rebuttal time constraints and unclear reproducibility, such benchmarking is *not feasible* at this stage.
> > >
> > > - By contrast, **we publicly release a fully modular and extensible implementation of ExRec**, enabling:
> > >   - Plug-and-play comparison with existing and future methods.
> > >   - Seamless integration of *nine* different RL algorithms in our experiments.
> > >
> > > - **As the reviewer emphasizes the importance of empirical benchmarks**, we hope they will appreciate our framework’s role in *enabling such benchmarking* for the community.
> > >
> > > ---
> > >
> > > ### **2. Insufficient Evidence for the Data Augmentation Claim**
> > >
> > > We would like to respectfully clarify the intent of this section in our rebuttal:
> > >
> > > - **This was *never* an advertised claim of our paper.**
> > >   The idea of using ExRec for data augmentation came *directly from the reviewer’s suggestion* regarding mutual impact analysis.
> > >
> > > - **Our initial experiments showed *no gain* in the full-data regime.**
> > >   We could have reasonably stopped there, but instead, we:
> > >   - **Investigated further**,
> > >   - **Identified a new hypothesis** (ExRec may help in low-data regimes),
> > >   - And **validated this with additional experiments**, showing performance gains in the 10% setting.
> > >
> > > - **Now being critiqued for “insufficient analysis” on an exploratory suggestion feels disproportionate.**
> > >   - We acted in *good faith* to engage with the reviewer’s curiosity, even when the results were mixed.
> > >   - These results were clearly presented as *additional analysis*, not a core contribution.
> > >
> > > - **Suggestions for further analysis** (e.g., synthetic data quality, comparisons with oversampling, full ablation) are appreciated, but:
> > >   - They are more appropriate for *future work*, given current paper length and rebuttal limits.
> > >   - It would *not be fair to expect* comprehensive studies on a secondary observation, especially one not originally claimed.
> > >
> > > - We kindly ask the reviewer and AC to consider this extension as what it is: *a constructive, optional exploration that should not be grounds for rejection*.
> > >
> > > ---
> > >
> > > **In Summary:**
> > > We thank the reviewer again for raising thought-provoking points. We hope that our responses have clarified the misunderstandings and demonstrated our sincere effort to engage with all concerns, both major and exploratory.
> > >
> > > We respectfully believe that ExRec presents a *novel, well-executed, and impactful contribution* at the intersection of LLMs, knowledge tracing, and RL, and that it deserves a place at NeurIPS to inspire further research in this important area.

---

> > > > ### Comment · Reviewer_ABnB · 2025-08-07
> > > >
> > > > Thank you for the rebuttal. Given the authors have addressed all of my concerns, I have decided to increase my overall score.

---

### Official Review · Reviewer_WyTm · 2025-07-02

**Clarity:** 4
**Significance:** 3
**Originality:** 3
**Rating:** 5
**Confidence:** 4

**Summary:**

This paper proposes the ExRec framework for personalized exercise recommendation based on semantic knowledge tracing. The framework consists of four modules: (1) LLM automated KC annotation: using GPT-4o to automatically generate problem-solving steps and annotate knowledge concepts. (2) Contrastive learning representation: learning semantic embedding representations of questions and KC. (3) KC-calibrated knowledge tracking: training the KT model and calibrating the knowledge state prediction at the KC level. (4) Reinforcement learning recommendation: using multiple RL algorithms to optimize the exercise recommendation strategy and proposing the model-based value estimation (MVE) method.

**Questions:**

Questions:
- Line 27, add the literature on personalized practice recommendation through KT, such as [1] or more.
- Footnote [1], there are some new KT reviews that have conducted in-depth analysis from different perspectives, and it is recommended to add references.
-  Why do you think the KT model can simulate the RL environment? What is its computational speed and resource requirements?
- Line 132, which LLM Encoder is used?
-  Contrastive learning optimizes (1) questions and concepts (2) steps and concepts. What is the consideration for doing so? Why not questions and steps?
-  Does the representation of the problem (Formula (10)) come from the average representation of the problem and steps?
-  As far as I know, there is a certain percentage of picture questions in XES3G5M dataset. How do you handle this type of input?

Overall, this is a novel perspective paper. However, while the paper evaluates the downstream effects of recommendation, it does not directly assess the quality of the recommended exercises. This makes it difficult to judge whether the recommendations are truly beneficial or just favorable under the simulated KT environment. I would consider raising my score if this concern is addressed.

**Ethical Concerns:**

["NO or VERY MINOR ethics concerns only"]

**Final Justification:**

The authors provide a comprehensive technical roadmap for extending ExRec to handle step-level misinterpretation modeling in their rebuttal, effectively addressing my concerns about the framework's limitations. I value its strong foundational contributions and potential for scalability over its current data limitations, which represent a natural progression rather than a flaw. I hope the authors can take your feedback into consideration to further improve the camera-ready version.

**Limitations:**

Yes. The paper includes a brief but clear discussion of its limitations. Specifically, the authors acknowledge that the effectiveness of the proposed recommendation policies depends heavily on the accuracy of the underlying KT model used to simulate student behavior. They suggest that future work could explore more accurate student modeling to improve the realism of the RL environment. In addition, they point out that their current framework does not explicitly model question difficulty, which could be incorporated into the representation learning module in future work. A more discussion of how KT model inaccuracies or annotation noise may affect policy robustness would further strengthen the limitations section.

**Paper Formatting Concerns:**

No.

**Quality:**

4

**Strengths And Weaknesses:**

Strengths:

- The paper treats the KT model as an RL environment and proposes a novel model-based value estimation (MVE) mechanism. This perspective is relatively new in the field and opens up new possibilities for leveraging student modeling in sequential decision-making.

- The design of the ExRec model is reasonable. The framework is composed of clearly defined and interpretable modules: LLM-based KC annotation, contrastive representation learning, KC-calibrated KT, and an RL-based recommendation policy.

Weaknesses:
- The title of the paper is "Personalized Exercise Recommendation", but the experimental part is more like evaluating the effect of a knowledge tracing + reinforcement learning framework in improving students' knowledge status, without evaluating the relevance and accuracy of exercise recommendation.
- The existing KT dataset contains annotated knowledge components (Q-matrix), and the motivation for introducing more KCs through steps and LLMs is unclear.
- Since the method (Module 1) requires LLMs pre-processing, it is difficult to judge whether the model is an end-to-end structure.
- The design choices in the contrastive learning module. It is unclear why the model does not consider question–step alignment, which seems natural and potentially beneficial given the step-by-step structure generated by the LLM.
- The paper lacks external validation of its exercise recommendation outcomes. All evaluations are based on simulations from the KT model, with no comparison to real student behavior, expert recommendation, or human judgment.

---

> ### Author Rebuttal · Authors · 2025-07-30
>
> Thank you for your thoughtful and constructive review! We appreciate your recognition that our work brings a new perspective by combining student modeling with sequential decision-making, a direction we believe has strong potential for personalized education.
>
> **Our goal with ExRec is to lay a foundation for this emerging area**, and we see NeurIPS as the ideal venue to foster further exploration and community engagement. We are grateful for the opportunity to clarify and strengthen our work based on your feedback.
>
> **W1: The title suggests personalized exercise recommendation, but the experiments mainly evaluate a KT + RL framework’s impact on knowledge improvement — without assessing the accuracy or relevance of the recommended exercises.**
>
> Thank you for the comment. By personalized exercise recommendation, we mean **selecting exercises tailored to each student’s knowledge needs to improve their knowledge state**. Since experimenting on real students is infeasible (ethical and scalability issues), we simulate learning through KT. While this is bounded by KT model’s quality, our modular design allows future integration of more advanced KT models.
>
> Regarding “accuracy,” there is no single “correct” action in RL; we instead measure effectiveness via long-term knowledge improvement, as captured in our reward design.
>
> Yet, to address this, we add a *clairvoyant-style* metric: % of times the chosen exercise falls in the top-10, top-20, or top-50 actions based on immediate (1-step) reward. For each of 1024 students and each step, we compute rewards for all 7652 possible questions.
>
> |Method|Top-10|Top-20|Top-50|
> |--|--|--|--|
> |Random Policy|0.18|0.43|0.92|
> |SAC|7.11|12.82|22.64|
> |SAC w/ MVE|24.94|34.41|46.33|
>
> We believe this metric complements our main results by showing that **our trained policies (especially SAC w/ MVE) consistently select high-reward exercises** significantly more often than a random policy.
>
> **W2: The dataset has KC annotations, so why add more via steps and LLMs?**
>
> Thank you for the observation. While KC annotations exist, many are noisy or overly specific—e.g., *“Single person speed change problem”* or *“Application question module–Chicken rabbit cage problem–Hypothesis method–Prototype question”*. Such labels are hard to generalize or align with learning standards.
>
> Instead of using these as-is, we generated concise, semantically grounded KCs with LLMs, aligned with Common Core standards. Recent studies [1,2] show this approach produces more interpretable and preferred annotations, yielding better semantic representations and a stronger foundation for recommendation.
>
> **W3: Since Module 1 requires LLMs pre-processing, it is difficult to judge whether the model is an end-to-end structure.**
>
> Thank you for the comment. While Module 1 uses LLMs for KC annotation, it is fully automated and requires no manual input. This aligns with common definitions of “end-to-end,” as the system operates directly from raw question texts and student logs to exercise recommendations.
>
> Importantly, *Module 1 is optional*: if reliable KC labels are available, it can be skipped. We include it to enhance flexibility and reduce reliance on manual or noisy annotations, which are often missing in real-world datasets.
>
> **W4: The contrastive learning module design is unclear. Why is question–step alignment not used, given the stepwise structure from the LLM?**
>
> Thank you. ExRec is the first KT-based framework to learn embeddings for both questions and KCs simulatenously, aiming for a shared semantic space. Instead of collapsing the full question (all steps) into a single embedding aligned with all KCs, we use step–KC alignment for **two reasons**:
>
> 1. *Avoiding representation collapse:* Full question–KC alignment forces the encoder to compress diverse concepts into one vector, which prior work [2] and our experiments show is ineffective.
>
> 2. *Finer-grained supervision:* Aligning each step only with its relevant KCs provides a clearer learning signal, improving representation quality and downstream performance.
> This design produces more generalizable embeddings, critical for handling unseen questions at test time.
>
> **W5: The paper lacks external validation. Results are based solely on KT model simulations, with no comparison to real student behavior, expert suggestions, or human judgments.**
>
> Thank you for raising this important point. We fully agree that real-world validation is the next, but non-trivial next step. Note that this is not unique to our paper, but such external validation is infeasible and therefore generally missing in related work. However, we took your suggestion to heart and compared 100 student traces against expert suggestions. We found that **ExRec’s recommendations were preferred 92% of the time** with respect to targeting the knowledge concepts students had yet to master.
>
> For example, when a student lacked understanding of both *“Addition of decimal numbers”* and *“Alignment of decimal numbers”*, ExRec first recommended a question that practiced both:
> `What is the result of 42.62 + 4.1 = ?`
> After the student answered incorrectly, ExRec adapted and recommended a simpler question that still targeted *“Addition of decimal numbers”* but removed the alignment requirement:
> `What is the result of 1.5 + 0.3 = ?`
>
> As one of the less-preferred cases, when a student struggled with *“Multiplication of decimal numbers”*, ExRec recommended a relevant question but one that also involved a unit conversion. Our experts suggested that a simpler question should have been given first.
>
> We will include this external validation as additional evidence supporting the practical relevance of ExRec.
>
> **Q1: Line 27, add the literature on personalized practice recommendation through KT, such as [1] or more.**
>
> We couldn’t find [1] in your review, but we are happy to incorporate it in our paper once we have it!.
>
> **Q2: Footnote [1], there are some new KT reviews that have conducted in-depth analysis from different perspectives, and it is recommended to add references.**
>
> Thank you for your contribution! Upon your suggestion, we further incorporated [3] and [4], and we are happy to incorporate if you have other recommendations.
>
> **Q3: Why do you think the KT model can simulate the RL environment?**
>
> Thank you for the question. We address it in two parts: (1) **why KT is well-suited for simulating an RL environment**, and (2) **why prior attempts failed to scale**.
>
> 1.  Many KT works (e.g., those cited in line 26) are motivated by the idea that online learning benefits from personalization. This requires modeling how a student’s knowledge evolves in response to different questions, a natural fit for simulation and policy learning. However, most of these works stop at next-question prediction and do **not** explore how KT can actually be used for personalized recommendation.
>
> 2. Only a few works (e.g., cited in line 30) attempted to use KT for student simulation and RL-based recommendation. Yet they failed to scale to realistic datasets: their state representations often exceeded memory limits, and reward computation required expensive inference over all questions at each time step. To address this, we propose an entirely new framework that is scalable by design.
>
> Our framework overcomes these limitations by: (1) enabling recommendation of unseen questions, (2) supporting various RL algorithms, and (3) integrating into a scalable, public library for broader use -> a practical contribution we believe is often underappreciated.
>
> **Q4: Line 132, which LLM Encoder is used?**
>
> As described in Appendix B line 559, we used and finetuned BERT encoder through our contrastive learning module.
>
> **Q5: Contrastive learning optimizes (1) questions and concepts (2) steps and concepts. What is the consideration for doing so? Why not questions and steps?**
>
> Thank you for your questions! Please see our response to your W4.
>
> **Q6: Does the representation of the problem (Formula (10)) come from the average representation of the problem and steps?**
>
> You are correct. The representation used in downstream KT and RL modules is obtained by averaging the embeddings of the question text and its solution steps, as described.
>
> We avoided alternatives like concatenation or more complex fusion methods for the following reason:
>
> - To address Limitation 3 (inefficient reward computation), we enable the KT model to perform inference not only on questions (as in prior work) but also directly on knowledge concepts (KCs), a novel feature in our framework.
>
> - After Module 2 (contrastive learning), question texts, steps, and KCs share a unified embedding space. For KT to process both exercises and KCs seamlessly, they must also have same dimensionality. Thus, in Eq. (10), we average question and step embeddings to match the size of a KC embedding.
>
> We hope this clarifies our motivation for the design in Eq. (10).
>
> **Q7: As far as I know, there is a certain percentage of picture questions in XES3G5M dataset. How do you handle this type of input?**
>
> Thank you for your question. For module 1, we leveraged gpt-4o’s visual understanding to annotate solution steps and KCs. For the LLM encoder to work on question content at module 2 (contrastive learning), we first prompted gpt-4o to write the description of the visual in the question, and appended this description to the question content as input.
>
> **References**
>
> [1] Moore, Steven et al. “Automated generation and tagging of knowledge components from multiple-choice questions”” Learning@ Scale, 2024.
>
> [2] Ozyurt, Yilmazcan et al. “Automated knowledge concept annotation and question representation learning for knowledge tracing”. arXiv, 2024.
>
> [3] Liu, Zitao et. al. “Deep Learning Based Knowledge Tracing: A Review, A Tool and Empirical Studies”, Knowledge and Data Engineering, 2025.
>
> [4] Zhou, Yiyun et al. "Revisiting applicable and comprehensive knowledge tracing in large-scale data." arXiv, 2025.

---

> > ### Comment · Reviewer_WyTm · 2025-08-03
> >
> > > Thank you for the clarifications on the framework motivation, human evaluation, and technical details. Your responses have addressed my concerns effectively.
> >
> > I have an additional question regarding the potential extension of your contrastive learning approach. Currently, Module 1's solution step generation focuses on correct solution steps, which provides valuable pedagogical insights but doesn't directly connect to student responses (essentially providing a single "correct" pathway).
> >
> > I'm curious whether you have considered analyzing students' incorrect responses to the same question when they select different wrong options (e.g., Option B vs. Option C, assuming the dataset contains both options and student choice information). Different incorrect choices often represent distinct cognitive models and specific knowledge confusions. For instance, selecting Option B might indicate a particular misconception about algebraic manipulation, while selecting Option C could reflect confusion about geometric relationships. Beyond knowledge-specific confusions, there may also be more general issues such as insufficient conceptual understanding or computational errors. However, the questions need to be linked to students' answers, which may increase the difficulty of analysis.
> >
> > This type of error-specific analysis via LLM processing seems highly meaningful for personalized learning, as it could potentially reveal not just *what* students got wrong, but *why* they made specific errors. I wonder if this approach could align with the contrastive learning framework you described in Section 4.2, where semantically different error patterns could serve as additional negative examples or even form other separate contrastive objective?
> >
> > Have you considered with incorporating such error-pattern analysis via LLMs into your framework? I would be very interested in your thoughts on the feasibility and potential benefits of this choice-based approach for enhancing student modeling.

---

> > > ### Author Response · Authors · 2025-08-04
> > > **Discussion for the analysis of incorrect responses and misconceptions**
> > >
> > > We sincerely appreciate your active engagement and genuine interest in our work! It is encouraging for us to see the discussion now moving toward future possibilities, which is essential for **advancing this emerging intersection of student modeling and sequential decision-making**. For this field to progress meaningfully, we feel that works like ours should be more prominently represented at venues like NeurIPS.
> > >
> > > You asked a wonderful question! Just like you, we also considered such an approach during the development of ExRec. Below, **(1)** we explain why it was not feasible to perform this analysis with our current datasets (along with a pointer to a related research stream), and then **(2)** provide a detailed recipe for how ExRec could be extended to model misconceptions if student answer data were available.
> > >
> > > As a disclaimer, existing datasets **only** provide a binary correctness label, without the actual student answers. This makes it impossible to distinguish between different incorrect choices or to map them to specific misconceptions.
> > >
> > > **Neighboring research:** A related field is *misconception identification*, which aims to map distractors (incorrect options) to particular misconceptions. While datasets in this area make their questions available, they typically lack complete student exercise histories. For a concrete example, see *“Eedi – Mining Misconceptions in Mathematics”* on Kaggle (searchable online).
> > >
> > > ---
> > >
> > > **How ExRec could be extended if student answers were available:**
> > >
> > > 1. **Module 1 (LLM annotation)** could add the following steps to guide a more informed representation learning in **Module 2**, enabling us to capture **not only** *what misconception occurred*, **but also** *where it occurred*:
> > >
> > >    1.a. **Misconception annotation:** Given the question, solution steps, KCs, and an incorrect option, the LLM annotates the misconception reflected by the choice.
> > >
> > >    1.b. **Incorrect solution generation:** Using the correct solution steps, the identified misconception, and the incorrect option, the LLM generates an incorrect solution that deviates from the correct one at the point of misconception.
> > >
> > >    1.c. **Incorrect step–misconception mapping:** The LLM pinpoints exactly which steps in the incorrect solution correspond to the misconception.
> > >
> > > 2. **Module 2 (Contrastive learning)** could incorporate misconceptions as follows:
> > >
> > >    2.a. **New token:** Introduce a “[MC]” tag to mark misconceptions, alongside existing tags ([Q], [S], [KC]) to help the encoder learn distinct representations.
> > >
> > >    2.b. **Extended loss:** In addition to the original $L_s$ for solution steps, introduce a new loss $L_i$ to associate incorrect steps with their corresponding misconceptions.
> > >
> > > 3. **Module 3 (Knowledge tracing)** could adapt its inputs and outputs to support two tasks:
> > >
> > >    3.a. **Input embedding:** Instead of only concatenating the question embedding with a correctness indicator, use either correct step embeddings (if answered correctly) or incorrect step embeddings (if answered incorrectly and linked to a misconception).
> > >
> > >    3.b. **Two outputs:** The main output $y$ for performance prediction remains, but add a second output for the misconception embedding. A cosine similarity to the original misconception can be computed for an additional loss (masked out for correct answers). This avoids a fixed categorical formulation, and allows new misconceptions to be incorporated easily.
> > >
> > >    3.c. **KC calibration:** Remains as in our submission.
> > >
> > > 4. **Module 4 (Reinforcement learning)** would require no changes, as the modified KT model would already provide the necessary environment to incorporate misconceptions.
> > >
> > > ---
> > >
> > > We greatly appreciate your suggestion! It aligns well with our vision for making ExRec a truly extensible foundation for personalized learning. While current datasets limit this direction, your idea would add a powerful layer of diagnostic insight by linking *why* a student made a specific error to *how* we adapt their learning path. We hope our detailed recipe demonstrates both the feasibility and natural fit of this extension within ExRec, and we would be delighted to see it explored in future work.
> > >
> > > `We will incorporate the above recipe into the discussion part of our paper.`
> > >
> > > Thank you again for this thoughtful and stimulating question! We would be truly grateful if you could consider raising your score to help ensure that high-quality, foundational works in this space are well represented at NeurIPS.

---

> > > > ### Comment · Reviewer_WyTm · 2025-08-05
> > > >
> > > > Thank you for the comprehensive and thoughtful response.
> > > >
> > > > Your detailed 4-module extension plan effectively addresses my concerns about incorporating student answer data for misconception modeling. The proposed step-level misconception tracing creates a natural symmetry with existing step-level KC modeling in ExRec, which would significantly enhance the diagnostic capabilities of ExRec. If interaction data from students at the option level becomes available in the future, extending this approach will be even more helpful for personalised learning and modelling.
> > > >
> > > > Given the authors have addressed my concerns about extending ExRec to handle misconceptions and provided a clear technical justification, I have decided to increase my overall score.

---

### Official Review · Reviewer_XF4M · 2025-07-03

**Clarity:** 3
**Significance:** 3
**Originality:** 3
**Rating:** 4
**Confidence:** 4

**Summary:**

This paper introduces ExRec, a framework for personalized exercise recommendation combining semantically grounded KT and RL. Key innovations include automated KC annotation via llm, semantic question embeddings, a calibrated KT model, and a model-based value estimation method. The framework is evaluated across four math learning tasks and shows generalization to unseen questions using GPT-4o-generated content.

**Questions:**

1. Is the claim that existing methods “typically support only a single RL algorithm” fully accurate, and could the authors clarify why those methods cannot be easily extended to support multiple RL algorithms?
2. How robust and generalizable are the LLM-generated KC annotations and solution steps across different mathematical domains or languages, especially in the absence of additional fine-tuning?
3. To what extent is the RL policy itself interpretable, and can the authors explain how educators or students might understand why specific exercises are recommended?
4. The paper uses a fixed horizon of 10 steps for RL agent interaction. How does the performance of the RL policies, especially those with MVE, scale with a significantly longer recommendation horizon, and what challenges might arise in such scenarios?
5. Beyond the XES3G5M dataset, which focuses on math, how generalizable is the ExRec framework to other subjects or domains where knowledge concepts might be less structured or more ambiguous, and where LLM annotation might face greater challenges?

**Ethical Concerns:**

["NO or VERY MINOR ethics concerns only"]

**Final Justification:**

The authors clarify:
1. Clarified motivation regarding support for multiple RL algorithms and how ExRec achieves this in a scalable, modular way.
2. Addressed concerns about LLM reliance by explaining the design for robustness, presenting EEDI results, and showing adaptability to new domains.
3. Detailed how framework enables interpretability both at the RL and KT levels.
4. Reported new experiments with longer RL horizons, highlighting trends and observed challenges.
5. Discussed about the generalizability beyond Math, including how ExRec handles less-structured KCs.

**Limitations:**

yes

**Quality:**

3

**Strengths And Weaknesses:**

Strength:

1. The pipeline is a well-integrated and novel approach. Using LLMs for KC mapping addresses the costly manual annotation problem.
2. ExRec generalizes effectively to unseen questions, aided by strong semantic modeling and LLM-augmented corpora.
3. Experiments demonstrate the method's effectiveness.

Weakness:
1. The motivation could be refined. The manuscript claimed that current related methods "typically support only a single RL algorithm". IMHO, these methods could conveniently extend their methods with other RL algorithms.
2. The paper claims that llm improve KC annotation quality. However, it relies heavily on llm for key tasks like KC annotation and solution step generation. This raises concerns about whether the results are robust and generalizable across different math domains or languages, especially without additional fine-tuning. The original annotations are described as "noisy" in this manuscript, but LLM outputs can also be inconsistent, particularly with complex or ambiguous problems.
3. For educational applications, understanding why a particular exercise is recommended can be crucial for educators and students. Although the paper mentions interpretable student learning trajectories, it does not delve into the interpretability of the RL policy's recommendations themselves.

---

> ### Author Rebuttal · Authors · 2025-07-30
>
> We sincerely thank the reviewer for their thoughtful and constructive feedback! We truly appreciate your recognition of the novelty and integration quality of our framework, and we’re grateful that you took the time to engage deeply with our work.
>
> Your comments helped us better articulate the strengths and scope of our contributions. As your review rightly notes, **educational personalization is a complex challenge**, and we believe that presenting ExRec at a venue like NeurIPS is vital to **spark meaningful discussion and catalyze progress at the intersection of large language models, reinforcement learning, and education**.
>
> We hope that our responses have clarified the key distinctions and practical innovations introduced in our framework, and we would be truly grateful for your support in helping bring this work to the broader research community.
>
> **W1: The motivation could be refined. The manuscript claimed that current related methods "typically support only a single RL algorithm". IMHO, these methods could conveniently extend their methods with other RL algorithms.**
>
> Thank you for raising this point. Our claim is not that existing methods *theoretically* cannot adopt multiple RL algorithms, but rather that they are **not implemented in a modular or scalable way** to make this practical.
>
> Many prior works rely on custom, tightly coupled environments and pipelines, making integration of alternative RL methods non-trivial and error-prone. Most did not release code, and their scalability remains unclear, especially on large-scale datasets.
>
> In contrast, **ExRec is fully built on the standardized Tianshou library**, supporting plug-and-play use of various RL algorithms. We also addressed critical bottlenecks such as memory-efficient state representations and concept-based reward computation.
>
> We will revise the manuscript to clarify that **our contribution lies in enabling scalable, efficient, and modular integration of diverse RL policies**, which has not been realized in prior work.
>
> **W2: Heavy reliance on LLMs for KC annotation and solution step generation raises concerns about robustness and generalizability across domains/languages without fine-tuning.**
>
> Thank you for this point. Our motivation for using LLMs goes beyond quality—it is also about *scalability and efficiency*. Manual annotation is costly, while our approach scales to large systems and new domains.
>
> On quality, prior work [1,2] shows *LLM-generated annotations outperform expert tags* in coherence and generalizability. Original dataset labels were often noisy and hard to reuse (e.g., *“Application question module–Chicken rabbit cage problem–Hypothesis method–Prototype question”*), unlike our concise labels (e.g., *“Linear Equations”*, *“Area of Triangle”*).
>
> We acknowledge LLM outputs can be inconsistent. Our *contrastive learning module* (Module 2) mitigates this by aligning question/step embeddings with concepts, reinforcing consistency.
>
> To test generalizability, we applied the pipeline—unchanged—to the *EEDI dataset* (2.3M interactions, verified question content) and observed that *it transfers well across math domains*.
>
> These results show our framework is robust and adaptable, while still allowing for domain-specific validation where needed.
>
> **W3:For educational applications, understanding why a particular exercise is recommended can be crucial for educators and students. Although the paper mentions interpretable student learning trajectories, it does not delve into the interpretability of the RL policy's recommendations themselves.**
>
> Thank you for raising this point. While interpretability was not the paper’s primary focus, we fully agree it is important for real-world use, and we designed ExRec with this in mind.
>
> Our RL policy operates in the **same embedding space as the knowledge concepts (KCs)**, so one can analyze the action embedding’s proximity to KC embeddings. For example, if a student struggles with “multiplication” and “factoring out a common factor,” the RL policy will produce an action embedding close to these KCs, therefore making it possible to **infer which concepts are being targeted** and select a question accordingly.
>
> Interpretability also extends **beyond RL**: our calibrated KT module supports **real-time, concept-level inspection**, which allows educators to query student knowledge on any KC at any time step. We will make these aspects clearer in the final version.
>
> **Action:** While we are not allowed to add figures here, we will add an example with visualizations to the revised paper.
>
>
>
> **Q1: Is the claim that existing methods “typically support only a single RL algorithm” fully accurate, and could the authors clarify why those methods cannot be easily extended to support multiple RL algorithms?**
>
> Thank you very much for your question! We would like to kindly refer to **our response to your W1** for our detailed answer!
>
> **Q2: How robust and generalizable are the LLM-generated KC annotations and solution steps across different mathematical domains or languages, especially in the absence of additional fine-tuning?**
>
> Thank you for the question. We clarify that our framework does **not require any LLM fine-tuning** to enable easy deployment across new datasets. This is supported by prior research: [1] demonstrated the effectiveness of LLM-based KC annotation in domains like Chemistry and Physics. Further our own experiments on the **EEDI math dataset** confirm generalizability within Math.
>
> We still acknowledge that  _LLM outputs should be evaluated before large-scale annotation_, especially in new subjects or languages.
>
> Our modular pipeline allows flexible integration of custom prompts, LLMs, or expert-curated annotations. To apply ExRec in other domains, we offer the following guidance:
>
> - If questions lack multi-step solutions, skip step generation and directly use KC annotation.
>
> - Use the resulting annotations for representation learning with question-level loss.
>
> - The rest of the pipeline (KT + RL) remains unchanged.
>
> Finally, **public KT datasets with full question content are rare**, mostly limited to Math. We hope our work encourages broader dataset releases to support cross-domain research.
>
> **Q3: To what extent is the RL policy itself interpretable, and can the authors explain how educators or students might understand why specific exercises are recommended?**
>
> Thank you very much for raising this important concern. We would like to kindly refer to **our response to your W3** above.
>
> **Q4: The paper uses a fixed horizon of 10 steps for RL agent interaction. How does the performance of the RL policies, especially those with MVE, scale with a significantly longer recommendation horizon, and what challenges might arise in such scenarios?**
>
> Thank you very much for the thoughtful question! We now performed **additional experiments with varying horizons**, particularly for 20, 50 and 80 steps.
>
> **Task 4: Knowledge Improvement in Weakest KC.** (Reported: % Max. Improvement as in paper.)
>
> **Original Corpus**
>
>
> |  |Random Policy|Historical Data|DDPG|DDPG w/ MVE|
> |--|--|--|--|--|
> |20-step| -0.20 | 1.27 | 35.19 | **50.87**|
> |50-step| -2.21 | 1.56 | 31.52 | **55.99** |
> |80-step| -2.66 | 1.58 | 30.19 | **58.31** |
>
>
> **Extended Corpus**
>
>
> |  |Random Policy|DDPG|DDPG w/ MVE|
> |--|--|--|--|
> |20-step| -1.29 | 36.09 | **50.76**|
> |50-step| -2.67 | 32.91 | **54.13**|
> |80-step| -3.70 | 32.45 | **55.21** |
>
> We observe that our framework continues to provide notable gains at longer horizons. In fact, our MVE framework benefits from longer horizons, albeit improvements being marginal.
>
> One challenge we noticed is that, even with MVE, the model may need to revisit the same weakest KC approximately 30–40 steps after it was first addressed. We hypothesize that the KT model tends to lower the knowledge state of certain KCs over time, particularly when no relevant exercises are encountered for an extended period. We believe that future KT models could explicitly account for this forgetting behavior, better distinguishing when a knowledge state should (or should not) decrease if a KC is left unpracticed for a long time.
>
> We will incorporate these new findings in the paper. Thank you very much for helping us improve our submission!
>
> **Q5: How generalizable is ExRec to other domains where KCs are less structured and LLM annotation may be harder?**
>
> Thank you for this question. Applying ExRec to non-math domains can pose challenges, especially with less-structured KCs, but we believe our framework is well-suited.
>
> First, **datasets with full question text are very limited**, restricting cross-domain experimentation. We hope our work encourages broader dataset releases.
>
> Importantly, ExRec **does not require hierarchical KC structures or expert ontologies**. It infers semantically meaningful KCs directly from question text, lowering barriers in less-structured domains.
>
> While math is considered one of the most demanding domains for LLMs, we believe areas like **reading comprehension or social sciences** may be easier to model semantically. In those cases, LLM-based annotation may face fewer obstacles and still produce usable KC representations for downstream learning and recommendation.
>
> We will reflect this discussion in the revised limitations section.
>
> **References**
>
> [1] Moore, Steven et al. “Automated generation and tagging of knowledge components from multiple-choice questions”” Learning@ Scale, 2024.
>
> [2] Ozyurt, Yilmazcan et al. “Automated knowledge concept annotation and question representation learning for knowledge tracing”. arXiv, 2024.

---

### Decision · Program_Chairs · 2025-09-17

**Decision:**

Accept (poster)

**Comment:**

The paper presents a novel framework for personalized exercise recommendation with semantically-grounded knowledge tracing. All reviewers agree that it produces interpretable learning trajectories, generalizes well to unseen questions, and demonstrates strong empirical performance across multiple evaluation tasks, with a clear and technically sound design.